# Custom tuning of Rieske oxygenase reactivity

Jiayi Tian [1], Jianxin Liu [1], Madison Knapp [1], Patrick H. Donnan [1], David G. Boggs [1] & Jennifer Bridwell-Rabb [1]✉

Rieske oxygenases use a Rieske-type [2Fe-2S] cluster and a mononuclear iron center to initiate a range of chemical transformations. However, few details exist regarding how this catalytic scaffold can be predictively tuned to catalyze divergent reactions. Therefore, in this work, using a combination of structural analyses, as well as substrate and rational protein-based engineering campaigns, we elucidate the architectural trends that govern catalytic outcome in the Rieske monooxygenase TsaM. We identify structural features that permit a substrate to be functionalized by TsaM and pinpoint active-site residues that can be targeted to manipulate reactivity. Exploiting these findings allowed for custom tuning of TsaM reactivity: substrates are identified that support divergent TsaM-catalyzed reactions and variants are created that exclusively catalyze dioxygenation or sequential monooxygenation chemistry. Importantly, we further leverage these trends to tune the reactivity of additional monooxygenase and dioxygenase enzymes, and thereby provide strategies to custom tune Rieske oxygenase reaction outcomes.

The more than 70,000 members of the Rieske non-heme iron oxygenase (Rieske oxygenase) class of enzymes catalyze integral site-, chemo-, and stereo-selective reactions in catabolic and anabolic pathways[1–5]. These reactions mainly include the addition of one (monooxygenation) or two (dioxygenation) oxygen atoms into a substrate, but variations of these reactions are also known to exist and be catalyzed by these enzymes[1–6]. For example, some Rieske oxygenases have been implicated in catalyzing sequential monooxygenation reactions and others have been shown to catalyze monooxygenation reactions that lead to dealkylation, desaturation, or C-C bond cleavage[1,3–5]. This diverse chemistry is performed on a range of substrates that vary in size and complexity, and typically targets C- and N-centers that differ in hybridization. Yet, despite this diversity in reactivity, structurally characterized Rieske oxygenases exist as heterohexameric ($\alpha_3\beta_3$ or $\alpha_3\alpha'_3$), homotrimeric ($\alpha_3$), or homohexameric ($\alpha_3\alpha_3$) complexes[7–26]. In these architectural arrangements, the catalytic α subunit contains a conserved set of residues that bind a Rieske-type [2Fe-2S] cluster and a mononuclear iron site. The Rieske cluster shuttles electrons to the iron center across the subunit interfaces of the α protomers to facilitate oxidative chemistry[2,3,27,28]. Pioneering studies

have revealed that Rieske oxygenase catalysis is not trivial: electrons need to be delivered from an external source and moved to the iron center, and both a substrate and molecule of oxygen ($O_2$) need to bind in the active site[29,30]. More specifically, for reductive activation of $O_2$, a substrate must be present in the active site, iron atoms in both the Rieske cluster and non-heme iron site must be in a reduced state, and iron must transition from a six-coordinate to five-coordinate geometry[2,3,27,28,31]. Recent studies have indicated that an Fe(III)-super-oxo species is likely the key oxidant that is used to facilitate the monooxygenation and dioxygenation reactions performed by salicylate 5-hydroxylase and benzoate 1,2-dioxygenase, respectively (Supplementary Fig. 1)[32–34]. However, to date, despite several investigations that indicate formation of the reactive Fe-based intermediate for catalysis is the rate limiting step of a Rieske oxygenase catalyzed reaction, the nature of the oxidizing species for most members of this enzyme class remains to be defined[35–37].

Likewise, the structural motifs that Rieske oxygenases employ to facilitate their different catalytic outcomes remain unclear. In working to determine how Rieske oxygenases use a common catalytic scaffold and set of metallocenters to facilitate different chemical outcomes,

[1]Department of Chemistry, University of Michigan, Ann Arbor, MI 48109, USA. ✉e-mail: jebridwe@umich.edu

several foundational studies performed on carbazole dioxygenase (CARDO), naphthalene dioxygenase (NDO), cumene dioxygenase (CDO), and nitrobenzene dioxygenase (NBDO) revealed that providing alternative substrates to an enzyme of interest can promote different chemistry[20,38–40]. For example, single active site variants of the $\alpha_3$ Rieske oxygenase CARDO catalyze increased amounts of lateral dioxygenation chemistry on a carbazole substrate, and form amplified amounts of a monooxygenated 9-hydroxyfluorene compound[30,41]. Structures of CARDO in the presence of these compounds revealed two likely parameters that contribute to the different observed reactivity: the substrate orientation and the size of the active site pocket[20]. Experiments on the $\alpha_3\beta_3$ Rieske oxygenases NDO, CDO, and NBDO, on the other hand, revealed that both wild-type and variant proteins oxidize a wide range of substrates with different selectivities and also promote different reaction outcomes[38–40,42]. However, along with CARDO, each of these $\alpha_3\beta_3$ catalyzed reactions are generally noted to form product mixtures[38–40,42]. Therefore, whereas these innovative investigations show that Rieske oxygenases can sample different catalytic outcomes, few class-wide details exist regarding how a Rieske oxygenase can be rationally tuned to facilitate different chemical reactions.

Thus, in this work, to pinpoint motifs, trends, or active site residues that are broadly used to control chemical outcome in this enzyme class, we focused on *p*-toluenesulfonate methylmonooxygenase (TsaM), a member of the less well studied $\alpha_3$ class of Rieske oxygenases (Fig. 1a, b). TsaM is proposed to natively catalyze a monooxygenation reaction on the methyl group of *p*-toluenesulfonate and 4-methylbenzoate with the aid of a reductase, TsaB (Fig. 1b)[43–45]. This TsaM-catalyzed reaction was used as a model system for studying Rieske oxygenase reactivity because, like other degradative enzymes, it exhibits broad substrate specificities and performs chemistry on compounds that can be readily purchased, properties that were anticipated to be important for exploring reactivity[44]. TsaM was also chosen for this work because it catalyzes a monooxygenation reaction that is reminiscent of the reactions catalyzed by the Rieske oxygenases toluene dioxygenase (TDO) and chlorophyll(ide) *a* oxygenase (CAO, Fig. 1c, d)[10,46,47]. However, these enzymes, despite sharing the same catalytic metal-locenter machinery as TsaM catalyze remarkably different reactions. TDO catalyzes a dioxygenation reaction on an aromatic ring, rather than on the methyl group, of a structurally similar toluene substrate (Fig. 1c)[10]. CAO, like TsaM, performs chemistry on the methyl group of its substrate but instead catalyzes two monooxygenation reactions that result in the consecutive formation of a mono-oxygenated, sequentially monooxygenated, and formylated species (Fig. 1d)[46,47].

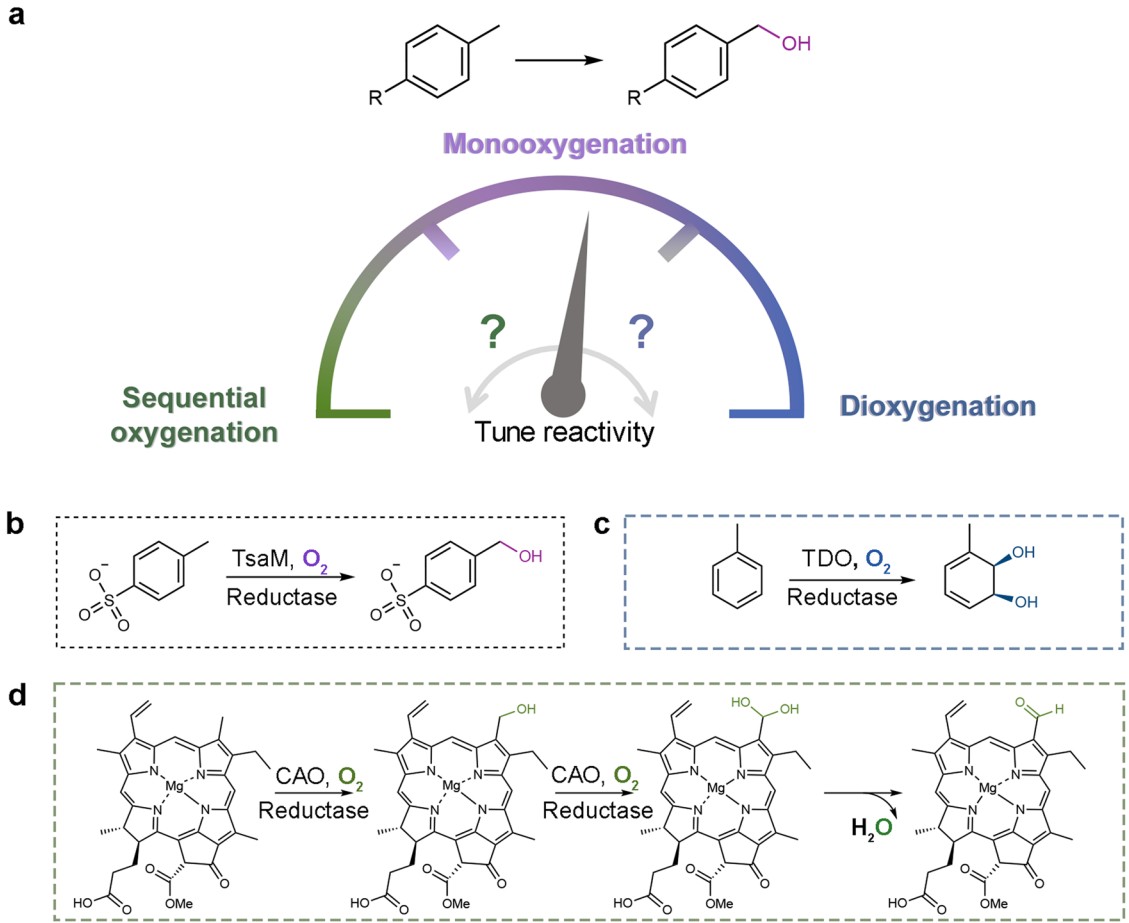

**Fig. 1 | TsaM is a model system for studying the architectural features that dictate reactivity in the Rieske oxygenase enzyme class. a** Here, it was investigated how the Rieske non-heme iron monooxygenase *p*-toluenesulfonate methyl-monooxygenase (TsaM) could be tuned to catalyze sequential monooxygenation or dioxygenation reactions. **b** TsaM, with the help of a reductase, is reported[43–45] to natively catalyze a monooxygenation reaction that transforms *p*-toluenesulfonate and 4-methylbenzoate into hydroxymethyl containing products. **c** This TsaM-catalyzed reaction is reminiscent of the reaction catalyzed by the Rieske oxygenase toluene dioxygenase (TDO) which instead catalyzes a dioxygenation reaction on an aromatic ring adjacent to a methyl group. **d** The TsaM-catalyzed reaction is also reminiscent of the reaction catalyzed by the Rieske oxygenase chlorophyll(ide) *a* oxygenase (CAO), which catalyzes sequential monooxygenation reactions on the C7-methyl group of a chlorophyll scaffold[47]. All Rieske oxygenase catalyzed reactions shown in this figure require a partner reductase protein to mediate the transfer of two electrons to the Rieske [2Fe-2S] cluster.

To identify the basis for different reactivity in this enzyme class, in this work, we use a combination of substrate and protein engineering experiments to manipulate TsaM to exclusively catalyze mono-oxygenation, dioxygenation, or sequential monooxygenation reactions (Fig. 1a). Collectively, the presented results suggest that the positioning of substrate in the active site of TsaM is guided by recognition elements on the substrate and establish that the distance between the substrate and mononuclear iron center is a key parameter for determining reaction outcome. Through additional work performed on vanillate *O*-demethylase (VanA) and phthalate dioxygenase (PDO), we reveal that the identified rational protein engineering strategy for tuning the reactivity of TsaM can be used to similarly manipulate the reaction outcome of other Rieske oxygenase enzymes: VanA is engineered to perform dioxygenation chemistry and PDO is engineered to function as a monooxygenase. Thus, this work identifies the basis for different reactivity in TsaM and pinpoints architectural trends that can be leveraged to support divergent chemical reactions. This information provides a framework for predictively changing the reactivity of Rieske oxygenases to perform specific oxidative transformations.

## Results

### Creation of an in vitro system for evaluating TsaM reactivity

A codon optimized gene that encodes TsaM from *Comamonas testosteroni*, which is also more formally known as *Pseudomonas testosteroni*, was synthesized with an N-terminal His-tag and Tobacco Etch Virus (TEV) protease cleavage site by Genscript. The expressed protein was purified as previously described[48] and assessed to be at least 90-percent pure using SDS-PAGE (Supplementary Fig. 2a). The presence of the Rieske cluster in TsaM was confirmed using a combination of UV-visible spectroscopy and iron analysis, and the ability of TsaM to form the characteristic trimeric architecture of a Rieske oxygenase was confirmed using gel-filtration chromatography (Supplementary Fig. 2b, c). Once purified, the activity of TsaM was tested using the hydrogen peroxide ($H_2O_2$) shunt reaction. This reaction bypasses the need for a partner reductase protein through direct formation of an activated $O_2$ intermediate using $H_2O_2$[49,50]. Combination of both [18]O-labeled and unlabeled $H_2O_2$ with TsaM and the reported native substrates[43–45], *p*-toluenesulfonate (**1**) and 4-methylbenzoate (**2**), revealed, using LC-MS, the expected hydroxymethyl-containing products (**3** and **4**, Fig. 2a).

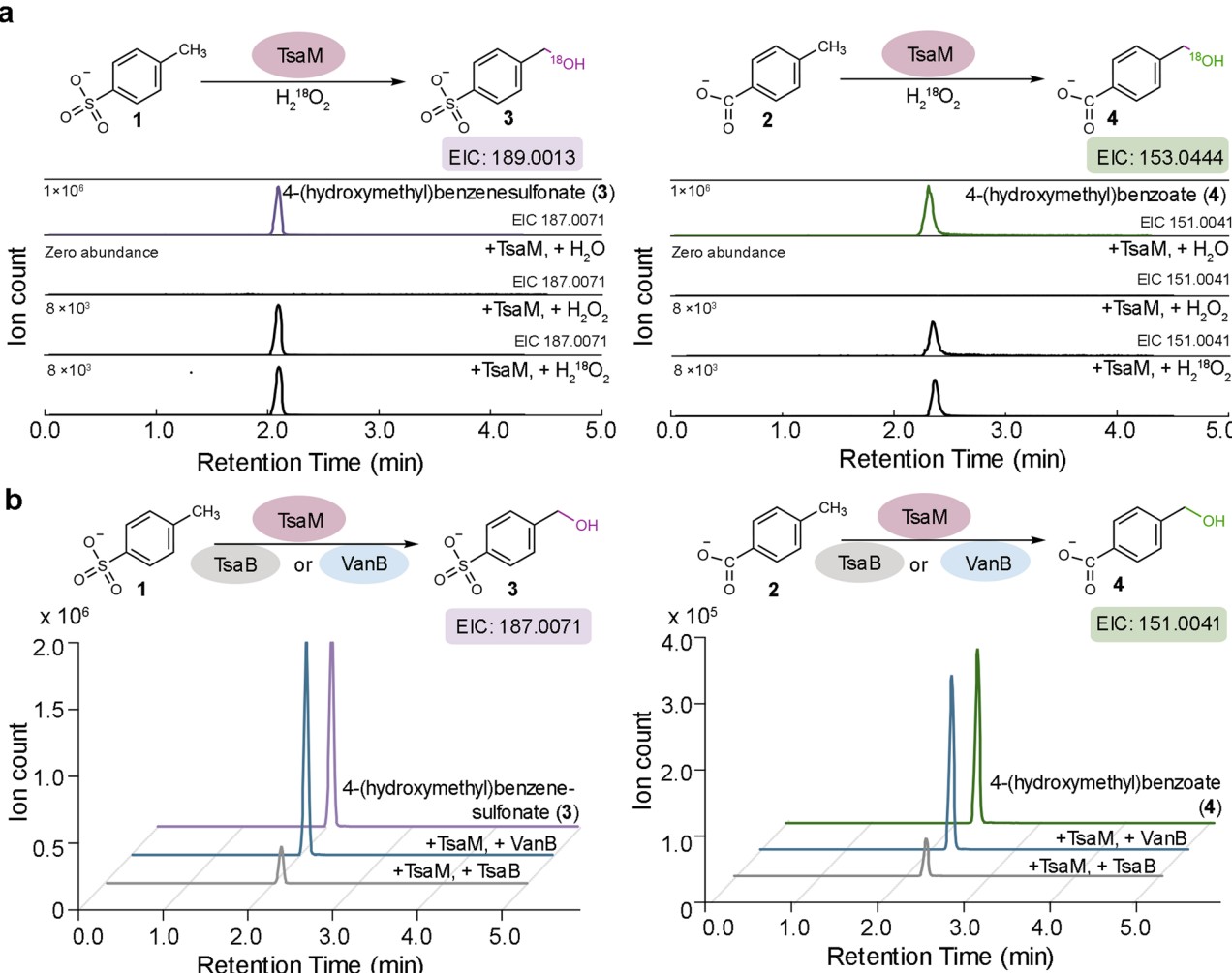

**Fig. 2 | TsaM catalyzes monooxygenation reactions to form hydroxymethyl containing products. a** The extracted ion chromatograms reveal that mono-oxygenated products, 4-(hydroxymethyl)benzenesulfonate (**3**, m/z = 187.0071 or 189.0013) and 4-(hydroxylmethyl)benzoate (**4**, m/z = 151.0041 or 153.0444) are formed when TsaM is provided with [18]O-labeled or unlabeled $H_2O_2$ and *p*-toluene-sulfonate (**1**) or 4-methylbenzoate (**2**), respectively. **b** Similar to that observed with $H_2O_2$, TsaM forms the expected products when combined with either TsaB or VanB.

Of note, more substantial amounts of 4-(hydroxymethyl)benzenesulfonate and 4-(hydroxylmethyl)benzoate are formed with each substrate when VanB, rather than TsaB, is included in the reactions. The data in panel b reflects results previously described by our laboratory[48]. Additional details for this panel regarding the measured total turnover numbers with the two different reductase proteins can be found in Supplementary Fig. 6.

**Table 1 | Summary of apparent kinetic parameters for TsaM-VanB with different substrates**

| Substrate | $K_M^{app}$ (µM) | $k_{cat}^{app}$ (min$^{-1}$) | $V_{max}$ (µM/min) | $k_{cat}^{app}/K_M^{app}$ (M$^{-1}$ sec$^{-1}$) |
|---|---|---|---|---|
| p-toluenesulfonate (**1**) | 3.8 ± 0.60 | 3.0 ± 0.22 | 6.2 ± 0.30 | 13000 ± 2200 |
| 4-methylbenzoate (**2**) | 13 ± 1.3 | 5.6 ± 0.21 | 11 ± 0.41 | 7200 ± 760 |
| p-aminotoluene (**5**) | 110 ± 7.7 | 1.6 ± 0.14 | 3.2 ± 0.23 | 230 ± 21 |
| p-nitrotoluene (**6**) | 230 ± 12 | 1.4 ± 0.25 | 2.9 ± 0.42 | 100 ± 16 |
| 3-methylbenzoate (**15**) | 100 ± 4.2 | 1.6 ± 0.37 | 3.1 ± 0.64 | 260 ± 50 |
| 4-ethylbenzoate (**19**) | 65 ± 3.8 | 3.6 ± 0.34 | 7.3 ± 0.62 | 930 ± 95 |
| 4-isopropylbenzoate (**25**) | 150 ± 6.5 | 2.1 ± 0.26 | 4.1 ± 0.40 | 230 ± 24 |
| p-(methoxy)benzoate (**31**) | 180 ± 8.9 | 1.6 ± 0.25 | 3.3 ± 0.42 | 150 ± 20 |
| p-(methylamino)benzoate (**33**) | 240 ± 11 | 1.6 ± 0.13 | 3.2 ± 0.29 | 110 ± 8.8 |
| p-(methylthio)benzoate (**34**) | 240 ± 2.5 | 1.3 ± 0.16 | 2.6 ± 0.41 | 88 ± 14 |
| benzenesulfonate (**37**) | 76 ± 3.6 | 0.58 ± 0.20 | 1.1 ± 0.40 | 120 ± 47 |
| benzoate (**38**) | 80 ± 1.5 | 0.52 ± 0.13 | 1.0 ± 0.22 | 99 ± 20 |
| 4-hydroxybenzenesulfonate (**41**) | 80 ± 2.8 | 0.66 ± 0.10 | 1.3 ± 0.21 | 130 ± 22 |
| 4-hydroxybenzoate (**32**) | 86 ± 2.1 | 0.50 ± 0.15 | 1.0 ± 0.25 | 98 ± 20 |

Following confirmation that the purified sample of TsaM was active, purification of the annotated native reductase TsaB was undertaken. TsaB is annotated as a ferredoxin-NAD$^+$ reductase (FNR) or an FNR$_C$-type Rieske reductase, based on the presence of N-terminal FAD and NAD$^+$ binding site signatures and a C-terminal [2Fe-2S] cluster-binding motif[51]. As described above for TsaM, the gene encoding *P. testosteroni* TsaB was synthesized by Genscript, recombinantly expressed, and purified using affinity chromatography (Supplementary Fig. 3). The purification of TsaB, however, as previously described[48], proved more challenging than that of TsaM, and subsequent activity measurements performed using a combination of TsaM, TsaB, and p-toluenesulfonate or 4-methylbenzoate resulted in production of only low levels of 4-(hydroxymethyl)benzenesulfonate and 4-(hydroxymethyl)benzoate (**3** and **4**, Fig. 2b and Supplementary Fig. 4). Therefore, an alternative FNR$_C$-type Rieske reductase, VanB, was purified and used in the assays (Supplementary Fig. 3a). As previously indicated[48], here, it was determined that combination of TsaM with VanB and p-toluenesulfonate or 4-methylbenzoate resulted in the formation of higher quantities of the expected products (Fig. 2b and Supplementary Figs. 5 and 6). In all cases, control reactions lacking TsaM or the reductase contained no detectable amounts of 4-(hydroxymethyl)benzenesulfonate and 4-(hydroxymethyl)benzoate (Supplementary Fig. 4 and 5). As a foundation for investigating the reactivity of the TsaM-VanB system and delineating reactivity with different substate molecules, a steady-state kinetic analysis was performed on the reported native substrates[43–45]. As previously described[16], in this work, for all substrates tested, we report the apparent kinetic parameters. This statement is due to the fact that the saturating concentrations of NADH and O$_2$ for the TsaM-VanB system are not measured in this work and they have they not before been determined. Here, it was determined that the apparent $k_{cat}$ of p-toluenesulfonate is similar to that measured with 4-methylbenzoate (Supplementary Figs. 7 and 8, Table 1). The measured $K_M$ of p-toluenesulfonate is also approximately three times lower than that measured with 4-methylbenzoate, meaning that the catalytic efficiency of the TsaM-VanB system with the sulfonated substrate is nearly two times higher than that measured with the carboxylated substrate (Table 1).

### Identification of key substrate features for functionalization by TsaM

Inspired by the different magnitudes of the TsaM-VanB system to oxygenate the methyl groups of p-toluenesulfonate and 4-methylbenzoate, a study was undertaken to determine the substrate specificity of TsaM (Fig. 3 and Supplementary Table 1). Toward this goal, the activity of TsaM was tested using VanB as an electron donor and p-aminotoluene (**5**), p-nitrotoluene (**6**), and p-isopropyltoluene (**7**) as substrates (Fig. 3a, Supplementary Figs. 9–11, and Supplementary Table 1). Here, using LC-MS, it was found that TsaM produces both 4-amino- and 4-nitrobenzyl alcohol products (**8** and **9**) when provided with p-aminotoluene or p-nitrotoluene, respectively (Fig. 3a, Supplementary Figs. 9 and 10). In contrast, no observable formation of 4-isopropylbenzyl alcohol (**10**) or another hydroxylated product is detected when p-isopropyltoluene is used in the reaction (Fig. 3a, Supplementary Fig. 11). To quantitatively compare the activity of TsaM with these substrates, the apparent kinetic parameters were analyzed (Table 1). For p-aminotoluene and p-nitrotoluene, it was determined that the measured $k_{cat}$ values are relatively similar (1.6 ± 0.14 and 1.4 ± 0.25 min$^{-1}$, respectively) but nearly four-times lower than that measured with 4-methylbenzoate. In addition, it was found that the $K_M$ for both compounds is approximately 8–60 times greater than that measured with p-toluenesulfonate and 4-methylbenzoate (Table 1). These results suggest that the sulfonate and carboxylate moieties of the substrate support enhanced catalytic activity of TsaM. Indeed, through the use of an AlphaFold model[52,53] of TsaM, a snapshot of the closest structurally characterized homolog, dicamba monooxygenase (DdmC)[24], and prior knowledge regarding important catalytic residues in TsaM[48], a potential substrate binding site was identified in TsaM (Fig. 3c). DdmC shares approximately 35-percent sequence identity with TsaM and contains a triad of residues that sit on one side of the active site and interact with the carboxylate moiety of dicamba (Asn230, His257, and Tyr263). Similarly, in the model of the TsaM active site, residues His255, Ser257, and Tyr269 form a polar cleft on one side of the active site, suggesting that these residues are key to recognizing the sulfonate and carboxylate moieties of the substrate (Fig. 3c and Supplementary Fig. 12). Consistent with the importance of these polar and charged substrate functional groups for observing high levels of TsaM activity, assays performed with toluene (**11**), which lacks a charged functional group completely, show no production of benzyl alcohol (**12**, Fig. 3a, Supplementary Fig. 13). A similar lack of activity is also obtained when p-chlorotoluene (**13**) is provided as a substrate to the TsaM-VanB system, suggesting that the size and geometry of the functional group at the C1 position of the substrate may also be important for functionalization (Fig. 3a, Supplementary Fig. 14).

To further explore the vastly different activities observed with this range of substrates (**1, 2, 5, 6, 7, 11,** and **13**), a previously described[54–56] assay was employed, with small deviations, to detect H$_2$O$_2$ formation. This colorimetric assay was implemented to determine whether incubation of TsaM-VanB with the different tested substrates resulted in

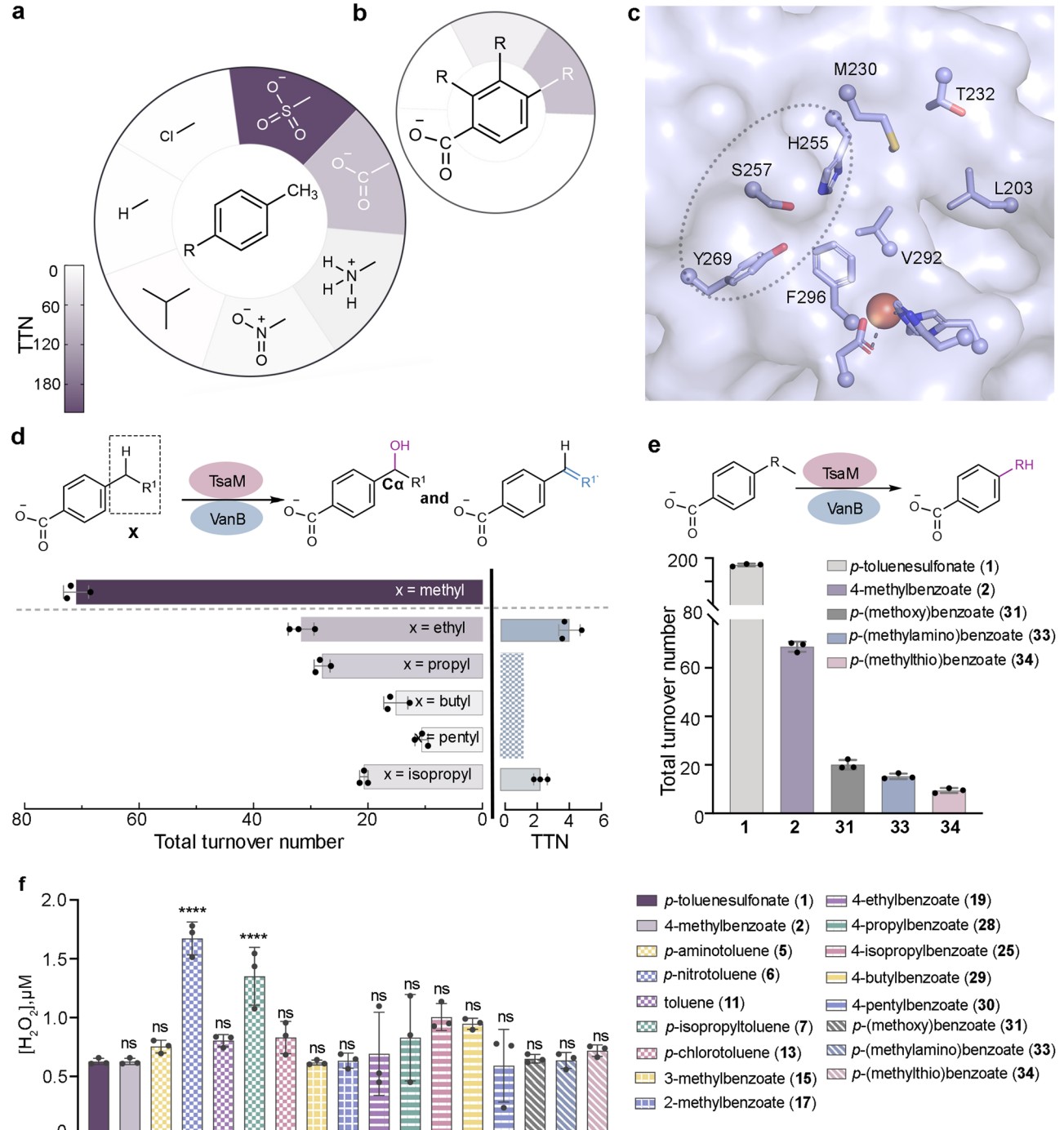

**Fig. 3 | TsaM demonstrates a preference for oxygenating the methyl group or benzylic carbon of substrates that contain a polar functional group. a** TsaM oxygenates *p*-toluenesulfonate (**1**), 4-methylbenzoate (**2**), *p*-aminotoluene (**5**), and *p*-nitrotoluene (**6**). In contrast, substrates that lack a charged functional group at the *p*-position (*p*-isopropyltoluene, **7**, toluene, **11** and *p*-chlorotoluene, **13**) are not functionalized by TsaM. The reported[43–45] native C1 substrate functional groups are indicated in white. **b** TsaM accepts and oxygenates 3-methylbenzoate (**15**) but does not oxygenate 2-methylbenzoate (**17**). This panel reflects previously described results[48], and, like panel a, also shows activity as a heat map with the native *p*-position of oxygenation in white. **c** The AlphaFold[52,53] model of TsaM[48], which is visualized using PyMOL 2.5.2_93 software, highlights a putative binding site for substrate (dashed circle). **d** Providing a 4-ethylbenzoate (**19**) or 4-isopropylbenzoate (**25**) to TsaM results in production of monooxygenated and desaturated products. The monooxygenated products are 4-(1-hydroxyethyl) benzoate (**20**) and 4-(2-hydroxy-2-propyl) benzoate (**26**), respectively. Total turn-over numbers (TTN) were also measured for TsaM with a 4-propyl- (**28**), 4-butyl-

(**29**), and 4-pentylbenzoate (**30**). These reactions also showed formation of both oxygenated and desaturated products, but the desaturated products were not quantified. TTNs are colored as described in panel **a**. **e** As previously described for *p*-(methoxy)benzoate[48] (**31**), TsaM catalyzes oxidative dealkylation chemistry when provided with *p*-(methylamino)benzoate (**33**) and *p*-(methylthio)benzoate (**34**). **f** A plot of the amount of H₂O₂ generated in the TsaM-VanB catalyzed reactions reveals a significant increase in H₂O₂ formation when *p*-nitrotoluene and *p*-iso-propyltoluene are provided as substrates to TsaM-VanB. In this panel ****$p < 0.0001$ and ns indicates no significant difference from an ordinary one-way ANOVA Tukey analysis. *P* values from left to right >0.9999, 0.9949, <0.0001, 0.8859, <0.0001, 0.7562, >0.9999, >0.9999, >0.9999, 0.7595, 0.0614, 0.1727, >0.9999, >0.9999, >0.9999, and >0.9999. In all panels, data was measured using $n = 3$ independent experiments and is presented as the mean value of these measurements. In panels **d**–**f** data are presented as mean values ± SD. For panels **a**–**c** and **f** additional details are provided in Supplementary Figs. 12, 15, and 17. Source data are provided as a Source Data file.

uncoupling of $O_2$ activation from substrate functionalization (Fig. 3f and Supplementary Fig. 15). This uncoupling phenomenon is known to result in lower product formation because the activated $O_2$ species is lost as $H_2O_2$ or as a different type of reactive oxygen species (ROS)[55,56]. The detrimental effect of uncoupling on enzyme activity can also be exacerbated by the ability of ROS to modify and inactivate an enzyme[55,56]. Interestingly, this investigation showed that incubation of p-aminotoluene, toluene, or p-chlorotoluene with TsaM does not result in an increased amount of uncoupling relative to that observed for TsaM with p-toluenesulfonate or 4-methylbenzoate. In contrast, combination of p-nitrotoluene or p-isopropyltoluene with TsaM results in significantly higher amounts of $H_2O_2$ formation, albeit with no detectable ROS-mediated protein modification (Fig. 3f and Supplementary Figs. 15 and 16).

In tandem with the importance of the functional group at the C1 position of the substrate for observing high levels of product formation, to build on prior results that indicate the position of the substrate methyl group also influences the activity of TsaM[48], we measured the apparent kinetic parameters for a 3-methylbenzoate substrate (15). Like that previously described[48], the use of a 3-methylbenzoate substrate in the assays results in lower formation of the corresponding monooxygenated product, 3-(hydroxymethyl)benzoate (16, Fig. 3b and Supplementary Fig. 17). This finding is accompanied by a decreased $k_{cat}$ and an increased $K_M$ relative to that observed with p-toluenesulfonate and 4-methylbenzoate (Table 1, Supplementary Table 1, and Supplementary Fig. 18). Since it was also shown that as previously indicated[48], movement of the methyl group to the ortho-position (2-methylbenzoate, 17) abolishes the ability of TsaM to catalyze an oxygenation reaction on the substrate, no kinetic parameters were able to be determined for this substrate (Fig. 3b, Supplementary Fig. 19). The ability of TsaM to oxygenate the secondary and tertiary carbon centers of substrates that contain an ethyl- or isopropyl- moiety in place of the methyl group on 4-methylbenzoate was also explored (Fig. 3d). Here, it was determined using LC-MS experiments, that when 4-ethylbenzoate (19) is given to TsaM as a substrate, as previously described[44], a monooxygenated product is formed (Supplementary Fig. 20). In our assays, like those previously performed[44], the identification of the reaction product is complicated by coelution of the proposed products, 4-(1-hydroxyethyl)benzoate (20) and 4-(2-hydroxyethyl)benzoate (21). Therefore, for product identification, the ability of Jones reagent to decipher between a primary and secondary alcohol via oxidation to the corresponding carboxylic acid and ketone products, respectively, was leveraged (22 and 23). In this experiment, Jones reagent, or a solution of 100 mM of $CrO_3$ in concentrated sulfuric acid, was combined with the enzymatic reaction mixture (Supplementary Fig. 21). Following a three-hour incubation at room temperature, it was observed that for the reactions treated with Jones reagent, a product with a mass, retention time, and MS/MS fragmentation pattern consistent with the ketone-containing product standard (23) was formed (Supplementary Fig. 21). Thus, it was concluded that TsaM reacts with 4-ethylbenzoate to form 4-(1-hydroxyethyl)benzoate. A steady state kinetic assay was performed to describe the reactivity of TsaM with this substrate. Here, it was found that the replacement of the methyl group on the substrate with an ethyl group, did not result in enhanced activity. Rather, the use of 4-ethylbenzoate as a substrate resulted in an approximate 7-fold decrease in the apparent catalytic efficiency of TsaM ($930 \pm 95$ $M^{-1}$ $sec^{-1}$) relative to that observed with 4-methylbenzoate (Table 1 and Supplementary Fig. 22). Of note, a small amount (produced in a 1:11 ratio) of a 4-vinylbenzoate (24) product was also identified using a commercially available standard (Fig. 3d and Supplementary Fig. 23). This minor product molecule is presumably produced via a desaturation reaction.

A compatible experiment was subsequently performed using TsaM, VanB, and 4-isopropylbenzoate (25, Fig. 3d and Supplementary Fig. 24). This reaction, which was hypothesized to have the potential to form either 4-(2-hydroxy-2-propyl)benzoate (26) or 4-(1-hydroxy-2-propyl)benzoate (27), also required implementation of the extra Jones oxidation step for product identification (Supplementary Fig. 25). Here, based on the lack of Jones oxidation product observed, it was determined that TsaM oxygenates this molecule with a preference for the benzylic carbon center to form 4-(2-hydroxy-2-propyl)benzoate. Using Michaelis-Menten kinetics, it was shown that the apparent catalytic efficiency, relative to the reported native substrates, is decreased, primarily attributable to an even higher $K_M$ ($150 \pm 6.5$ μM) than was measured in the reaction that contained 4-ethylbenzoate ($65 \pm 3.8$ μM, Table 1, Supplementary Fig. 26). Akin to the reaction with 4-ethylbenzoate, this reaction also revealed formation of a small amount of a desaturated product (Fig. 3d and Supplementary Fig. 27). To determine the length of sidechain that could be accepted by TsaM, additional comparisons to test the competence of 4-propyl-, 4-butyl-, and 4-pentylbenzoate as substrates of TsaM were performed (28, 29, and 30, Fig. 3d). As described for the product of the reaction using 4-ethylbenzoate, treatment of these products with Jones reagent yielded a ketone product, which suggests that the products of these reactions are also secondary alcohols (Supplementary Figs. 28–30). Combination of TsaM with these substrates also leads to formation of a small amount of desaturated product (Supplementary Figs. 28–30). Nevertheless, despite this additional reactivity, from these experiments a clear trend can be defined: increasing the substituent length at the para-position from a methyl to an ethyl, propyl, butyl, or pentyl functional group is correlated with lower total turnover numbers (Fig. 3d). The 4-isopropylbenzoate substrate is an outlier from this trend, presumably due to the branched, rather than linear, nature, of the isopropyl functional group (Fig. 3d). As described above, for each of these tested substrates, there is no significant difference in the amount of $O_2$ uncoupling relative to that observed with p-toluenesulfonate or 4-methylbenzoate (Fig. 3f).

With the knowledge that TsaM demonstrates a preference for hydroxylating the benzylic carbon when provided with 4-ethylbenzoate or 4-isopropylbenzoate, an additional investigation to complement previous reports[44,48] that TsaM accepts a p-(methoxy)benzoate (31) substrate in which the benzylic carbon atom is replaced with an oxygen atom, was performed (Supplementary Table 1). In this case, it was determined that, as previously described[48], combination of TsaM, VanB, and p-(methoxy)benzoate results in production of 4-hydroxybenzoate (32) and presumably formaldehyde (Fig. 3e and Supplementary Fig. 31a, b). Despite the structural resemblance of p-(methoxy)benzoate and 4-ethylbenzoate, it was found that the $K_M$ of TsaM is approximately 2.7-times higher with p-(methoxy)benzoate than was measured with 4-ethylbenzoate (Supplementary Fig. 31c, d). Similarly, when p-(methoxy)benzoate is substituted by a heteroatom containing functional group substrate, p-(methylamino)benzoate (33) or p-(methylthio)benzoate (34), in the reaction, 4-aminobenzoate (35) and 4-mercaptobenzoate (36) are produced, and the apparent $K_M$ values show even more pronounced increases (Table 1 and Supplementary Figs. 32–34). Despite the observed changes in the measured $K_M$ values when provided with these substrates relative to 4-ethylbenzoate the $k_{cat}$ values show only minor changes (Table 1 and Supplementary Figs. 32 and 33). Likewise, no significant increase in the amount of $H_2O_2$ formation, relative to TsaM with 4-ethylbenzoate, or the reported native substrates, was detected (Fig. 3f and Supplementary Fig. 15).

## Formation of a sequentially oxygenated product by TsaM
The impact of removing the methyl group from the substrate was also assessed. Here, it was determined that the combination of benzenesulfonate (37) or benzoate (38) with TsaM and VanB, results in formation of two dioxygenated aromatic products, 3,4-dihydroxybenzenesulfonate and 3,4-dihydroxybenzoate, respectively (39 and 40, Fig. 4a, Supplementary Table 1, and Supplementary Figs. 34

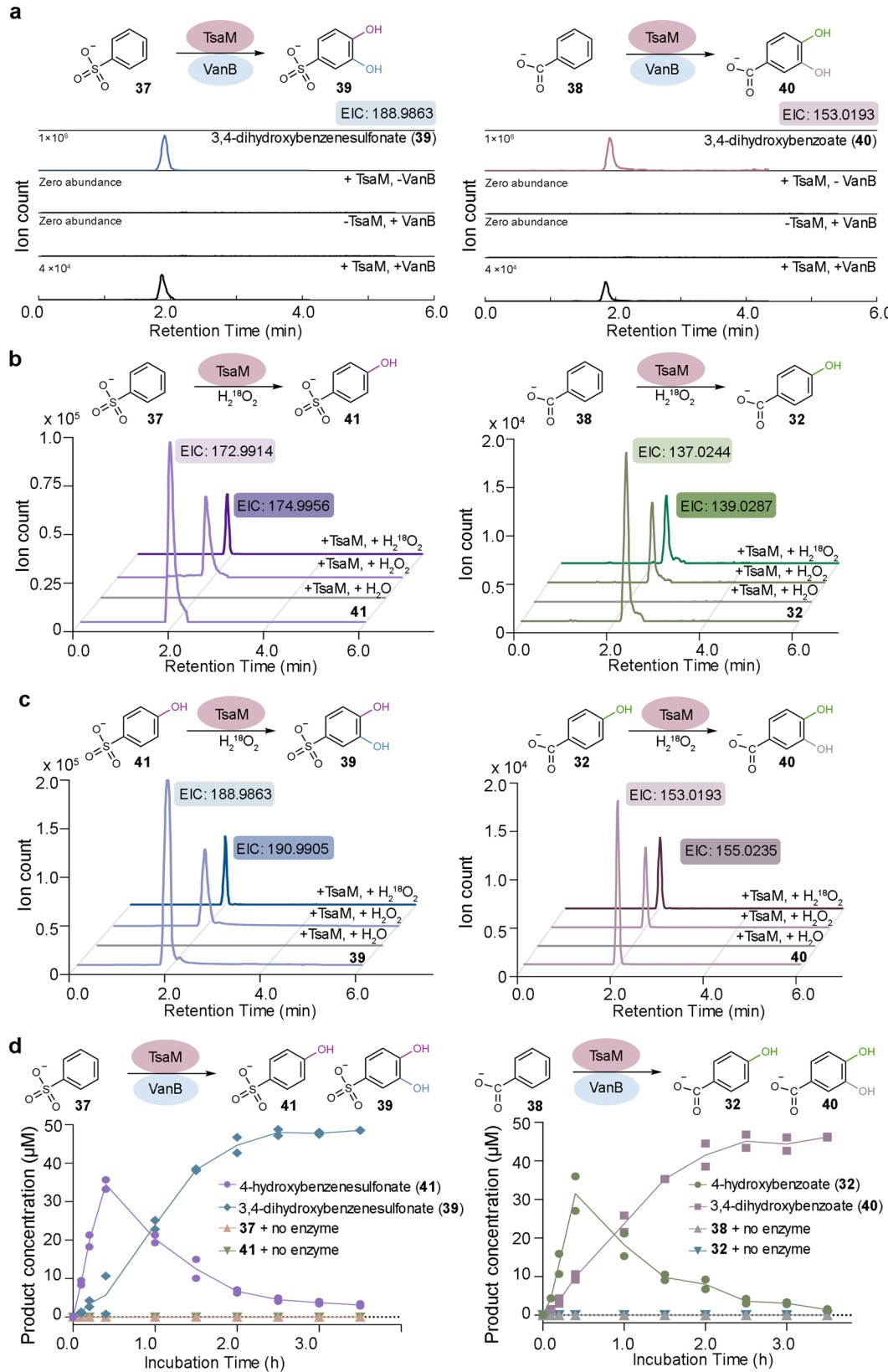

and 35). To determine whether these products were made via two iterative monooxygenation reactions or a single dioxygenation event, product formation was again tested using the $H_2O_2$ shunt reaction (Fig. 4b, c). This reaction was attractive for testing the mechanism of product formation because it permits assessment of a single enzyme turnover[49,50]. Here, it was determined that a single pass through the catalytic cycle using either a benzenesulfonate or benzoate substrate results in the formation of monooxygenated products, 4-hydroxybenzenesulfonate (**41**) and 4-hydroxybenzoate (**32**), respectively (Fig. 4b). Likewise, incubation

**Fig. 4 | TsaM produces a sequentially monooxygenated product when provided with a substrate that lacks a methyl group. a** Incubation of TsaM-VanB with benzenesulfonate (**37**) or benzoate (**38**) results in production of 3,4-dihydroxybenzenesulfonate (**39**) or 3,4-dihydroxybenzoate (**40**), respectively. Formation of these species can be accomplished either via sequential monooxygenation reactions or a single dioxygenation event. **b** Using the $H_2O_2$ shunt reaction, it was determined that incubation of benzenesulfonate with TsaM results in production of a monooxygenated compound, 4-hydroxybenzenesulfonate (**41**), after a single turnover. Analogous results are obtained when benzoate is included as the

substrate of the reaction. **c** Similarly, using the $H_2O_2$ shunt reaction, it was shown that 4-hydroxybenzoate (**32**) and 4-hydroxybenzenesulfonate can be transformed into 3,4-dihydroxybenzenesulfonate and 3,4-dihydroxybenzoate following an additional turnover, respectively. **d** Inspection of the TsaM-catalyzed reaction when provided with benzenesulfonate or benzoate reveals that first a monooxygenated species (purple and green) is produced and then a dioxygenated (light blue and pink) species is formed. In panel **d** the data was measured using $n = 2$ independent experiments. Additional details that pertain to panel **d** can be found in Supplementary Figs. 36–38. Source data are provided as a Source Data file.

of 4-hydroxybenzenesulfonate or 4-hydroxybenzoate with TsaM and VanB results in formation of 3,4-dihydroxybenzenesulfonate and 3,4-dihydroxybenzoate, respectively (**39** and **40**, Fig. 4c). These results are consistent with the observed products arising from two single monooxygenation events. Indeed, a schematic diagram of the reaction progress under multiple turnover conditions reveals an initial appearance of the monooxygenated products, 4-hydroxybenzenesulfonate and 4-hydroxybenzoate, consumption of these species, and formation of 3,4-dihydroxybenzenesulfonate and 3,4-dihydroxybenzoate (Fig. 4d and Supplementary Figs. 36 and 37). Based on the calculated total turnover numbers and apparent kinetic parameters for TsaM with benzenesulfonate, benzoate, 4-hydroxybenzenesulfonate, and 4-hydroxybenzoate, each of these substrates appears to support the same level of TsaM activity (Table 1 and Supplementary Figs. 35–38). Quite interestingly, despite these similarities, it appears that the initial rate of NADH consumption, in assays for both sets of substrates, is faster in the presence of the monooxygenated compounds (Supplementary Fig. 38g, h). A similar second oxygenation reaction on 4-hydroxybenzoate was also detected upon reexamination of the data for *p*-(methoxy)benzoate, which revealed the additional presence of 3,4-dihydroxybenzoate (Supplementary Fig. 39). Surprisingly, however, despite the sequential monooxygenation activity observed with benzenesulfonate, benzoate, and *p*-(methoxy)benzoate substrates, and the precedent for this type of chemistry to also happen on the C7 methyl group in the CAO catalyzed reaction[47], a similar ability to iteratively oxygenate the methyl group of *p*-toluenesulfonate or 4-methylbenzoate was not observed (Supplementary Fig. 40). As described below, we suggest that this inability is likely attributable to an architectural difference in the CAO active site that supports two rounds of the catalytic cycle.

Here, it is interesting to note that whereas TsaM catalyzes sequential monooxygenation reactions on the aromatic ring of benzenesulfonate and benzoate, equivalent reactivity is not observed with aniline (**42**) or phenol (**43**, Supplementary Figs. 41 and 42). This lack of activity and the low observed total turnover numbers for benzenesulfonate and benzoate substrates, relative to that observed with *p*-toluenesulfonate or 4-methylbenzoate substrates, again, prompted an investigation into whether the $O_2$ activation step had been uncoupled from substrate metabolism. Using the above-described assay, it was determined that using aniline, phenol, benzenesulfonate, or benzoate as a substrate of TsaM did not result in significantly higher formation of $H_2O_2$, relative to that observed with *p*-toluenesulfonate or 4-methylbenzoate (Supplementary Fig. 43).

## Identification of architectural parameters involved in TsaM reactivity

The observed ability of TsaM to catalyze both monooxygenation and sequential monooxygenation reactions laid the groundwork for investigating whether TsaM could also catalyze a dioxygenation reaction. To evaluate additional parameters, aside from substrate identity, that could contribute to the reaction catalyzed by TsaM, an analysis of the available substrate bound Rieske non-heme iron monooxygenase and dioxygenase structures was performed (Fig. 5a, Supplementary Fig. 44). Through this analysis, it was determined that, in these structures, the average distance between the iron center and the substrate

is approximately 0.7 Å shorter in the dioxygenases than in the monooxygenases (Fig. 5a). These findings inspired an exploration into how changing the distance between the iron center and the substrate would perturb the observed reactivity. To accomplish this task, *N*-phenylacetamide (**44**), phenyl acetate (**45**), or 4-methylphenyl acetate (**46**), which each contain a functional group that is longer than the carboxylate moiety of 4-methylbenzoate, were included in the enzymatic assays. However, when these molecules were provided as substrates to TsaM, no oxygenated products were formed (Supplementary Figs. 45–47). Rather, combination of either latter compound with TsaM results in a significant amount of $H_2O_2$ formation relative to that observed with *p*-toluenesulfonate and 4-methylbenzoate (Supplementary Fig. 43). As described above, with these substrates, we were unable to detect any protein-based modifications, suggesting that the lower levels of activity are likely only correlated with $H_2O_2$ formation (Supplementary Fig. 16). On the other hand, the use of benzoylformate (**47**) or 4-methylbenzoylformate (**48**) in the assay results in formation of both sequentially monooxygenated and singly oxygenated products (**49** and **50**), respectively, and no significant increase in $O_2$ uncoupling (Supplementary Figs. 43, 48 and 49).

As extending substrate size did not appear to eliminate the monooxygenation or sequential monooxygenation activity of TsaM, an investigation into how changing the size of the active site would perturb Rieske oxygenase reactivity was undertaken. For this experiment, the active site of DdmC was further inspected (Fig. 5b and Supplementary Fig. 50)[24]. This protein, as described above, packs its substrate in the active site between the carboxylate interacting residues (Asn230, His257, and Tyr263), and Leu202, Ile232, Gly255, Trp285, and Leu290 (Supplementary Figs. 12 and 50). Based on an alignment of the DdmC structure with the AlphaFold model[52,53] of TsaM, it was identified that Met230 and Thr232 are similarly positioned at the top of the TsaM active site reminiscent of DdmC residues Asn230 and Ile232 (Fig. 5b and Supplementary Fig. 50). To evaluate whether these residues could be used to push the substrate toward the iron center and impact the outcome of the TsaM reaction, Met230 and Thr232 were each mutated into bulky Phe and Trp residues (Supplementary Table 2).

For TsaM variants M230F, T232F, M230W, and T232W, which were shown to be properly folded using circular dichroism (CD) spectroscopy, it was determined that incubation with benzoate (**38**) and $H_2O_2$ did not result in the formation of a monooxygenated product as observed with the wild-type enzyme (Supplementary Figs. 51–54). Rather, this reaction yields two dioxygenated species, 3,4-dihydroxybenzoate (**40**) and a *cis*-dihydrodiol (3 *R*,4 *S*)-3,4-dihydroxycyclohexa-1,5-diene-1-carboxylate (**51**, Fig. 5c). The former molecule appears as a function of incubation time, presumably due to the conversion of (3*R*,4*S*)-3,4-dihydroxycyclohexa-1,5-diene-1-carboxylate into a more stable aromatic compound as previously described[57,58] (Supplementary Fig. 53). To probe the influence of a residue that is less bulky than Phe or Trp on the outcome of the hydroxylation event, a TsaM T232I variant was produced (Supplementary Table 2 and Supplementary Figs. 51, 52, and 54). This variant shows even lower production of the dioxygenated species, and unlike the Phe and Trp variants, did show some formation of the monooxygenated 4-hydroxybenzoate product (Fig. 5c and Supplementary Fig. 54). In contrast, a double M230W/

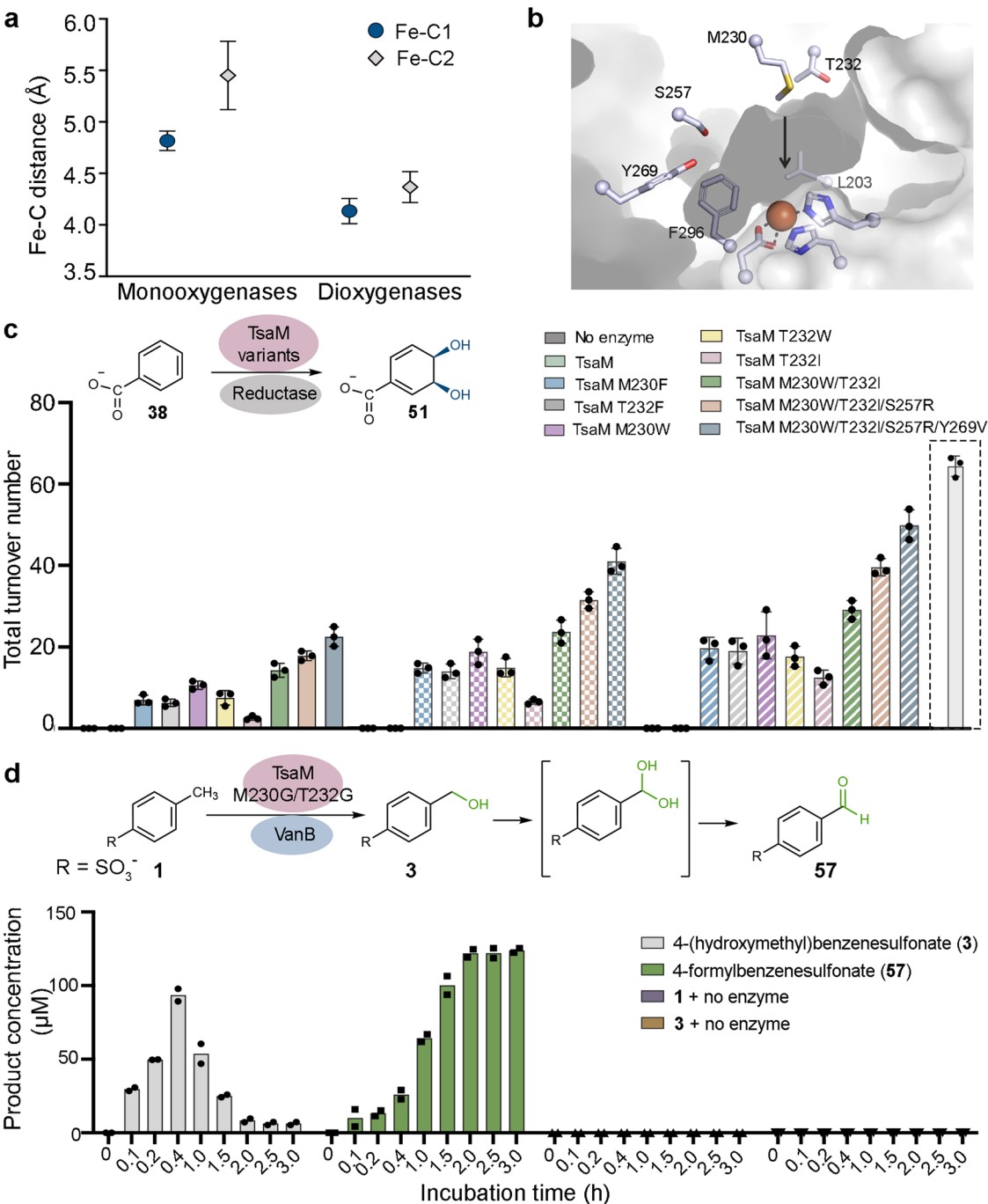

**Fig. 5 | A rational approach to change the reactivity of TsaM. a** The distances between the iron center and the two closest substrate carbon atoms (Fe-C1 and Fe-C2) were measured for available native substrate bound Rieske oxygenase structures. In this panel data was extracted from $n = 6$ (monooxygenase) and $n = 6$ (dioxygenase) independent structures. These values are presented as the mean value of the measured distances ±SD. This one-sided analysis reveals that the average distance between the iron and substrate is longer in the monooxygenases than in the dioxygenases ($p$ values for Fe-C1 and Fe-C2 distances = 0.0015 and 0.0207, respectively, as determined by an unequal variance t-test, see Supplementary Fig. 44). **b** The active site of dicamba monooxygenase (DdmC, PDB: 3GL2[24]) and an AlphaFold[52,53] model of TsaM[48], which are visualized using Pymol 2.5.2_93 software, were used to rationally engineer the TsaM active site. The Met230 and Thr232 residues in TsaM were mutated into Phe, Trp, Ile, or Gly residues to change the space between the iron and the substrate. **c** Using the TsaM M230F/W,

T232F/W, M230W/T232I, M230W/T232I/S257R, or M230W/T232I/S257R/Y269V variants, it was observed that TsaM exclusively produces a *cis*-dihydrodiol (**51**), rather than a monooxygenated species. T232I shows lower overall dioxygenation activity and produces some 4-hydroxybenzoate. From left to right, variant activity was measured in the presence of VanB (solid colors), PDR1 (checkered pattern), or PDR2 (striped pattern). The boxed bar is the activity of wild-type TsaM with a 4-methylbenzoate substrate. This data shows that the yield of the dioxygenation reaction approaches that of the reported native monooxygenation reaction with 4-methylbenzoate. In this panel data was measured using $n = 3$ independent experiments and are presented as mean values ± SD. **d** Providing the M230G/T232G variant of TsaM with *p*-toluenesulfonate (**1**) results in formation of a monooxygenated and sequentially monooxygenated (formylated) product over time (see Supplementary Fig. 63). In this panel, data was measured using $n = 2$ independent experiments. Source data are provided as a Source Data file.

T232I variant behaves similarly to M230W TsaM and forms only the dearomatized product (Fig. 5c, Supplementary Table 2, and Supplementary Fig. 54). Inspired by the above-described hypothesis that the carboxylate is positioned in the active site by His255, Ser257, and Tyr269, two final M230W/T232I/S257R and M230W/T232I/S257R/Y269V variants of TsaM were produced, purified, and shown to also exclusively create a *cis*-dihydrodiol species (**51**) when provided with a benzoate substrate (Supplementary Table 2 and Supplementary Figs. 51, 52 and 54). Remarkably, for these variants, the total turnover numbers for production of the dioxygenated compound are comparable to the total turnover numbers measured for the sequential monooxygenation reactions using benzenesulfonate and benzoate (Supplementary Fig. 38). This result suggests that the active site changes permit TsaM to catalyze a dioxygenation reaction with similar efficacy to the sequential monooxygenation reactions catalyzed by the wild-type enzyme (Fig. 5c and Supplementary Fig. 38). Importantly, it was determined that the latter, more active M230W/T232I/S257R and M230W/T232I/S257R/Y269V variants of TsaM still catalyze dioxygenation chemistry, albeit to a lower extent with the native reductase TsaB (Supplementary Fig. 54).

The identities of the dioxygenated products were verified in several ways: 3,4-dihydroxybenzoate was identified based on comparison of its mass, retention time, and MS/MS fragmentation pattern with a commercially purchased standard (Supplementary Fig. 55). The *cis*-dihydrodiol (**51**), on the other hand, was more challenging to characterize due to the lack of a commercially available standard. Therefore, to support the assignment of the reaction product as a *cis*-dihydrodiol, a standard of (3*R*,4*R*)-3,4-dihydroxycyclohexa-1,5-diene-1-carboxylate (**52**) was obtained. This molecule shows the same mass as the expected reaction product, (3*R*,4*S*)-3,4-dihydroxycyclohexa-1,5-diene-1-carboxylate, but is shifted in retention time, presumably due to its different atomic structure (Supplementary Fig. 56a). Nonetheless, this standard, like the dearomatized product (**51**), also converts into 3,4-dihydroxybenzoate over time (Supplementary Fig. 56b). Further, the product of the TsaM variant reactions and (3*R*,4*R*)-3,4-dihydroxycyclohexa-1,5-diene-1-carboxylate could both be converted into 3,4-dihydroxybenzoate by the addition of Pd/C as previously described[59] (Supplementary Fig. 57). As a final step to confirm that the product of the TsaM variants is a *cis*-dihydrodiol species, a second Rieske oxygenase, phthalate dioxygenase (PDO) from *P. testosteroni* KF1, was purified and biochemically characterized (Supplementary Fig. 58). Here, it was determined that combination of recombinantly produced PDO with VanB results in the expected transformation of phthalate (**53**) into phthalate *cis*-4,5-dihydrodiol (**54**, Supplementary Fig. 59). It was also shown that the PDO-VanB system catalyzes a dioxygenation reaction to transform benzoate into a molecule that has the same mass, retention time, and MS/MS fragmentation pattern as the *cis*-dihydrodiol product, again consistent with the assignment of the TsaM variant products as (3*R*,4*S*)-3,4-dihydroxycyclohexa-1,5-diene-1-carboxylate (Supplementary Figs. 60 and 61).

To rule out that the low observed turnover numbers for this engineered dioxygenase chemistry is due to O₂ uncoupling, an investigation into each of the tested TsaM variants (M230F/W, T232F/W/I, M230W/T232I, M230W/T232I/S257R, and M230W/T232I/S257R/Y269V) with benzoate did not reveal any significant increase in $H_2O_2$ formation relative to that observed in the wild-type system (Supplementary Fig. 62). As uncoupling did not appear to play a role in the low turnover of these variants, it was subsequently investigated whether the inclusion of an annotated dioxygenase reductase partner, rather than VanB, in the reactions, would amplify the amount of dioxygenated product formed. Here, it was determined that the use of either annotated PDO partner reductase from *P. testosteroni* (PDR1) or *Pseudomonas cepacia* (PDR2) in the reactions leads to an approximate doubling in the amount of the *cis*-dihydrodiol product formed by these variants (Fig. 5c and Supplementary Figs. 58 and 62). Quite interestingly, the amount of *cis*-dihydrodiol product formed by the M230W/T232I/S257R/Y269V TsaM variant in the presence of PDR2 is not far off from the amount of monooxygenated product formed by the TsaM-VanB system with 4-methylbenzoate, highlighting the marked amplification of activity imparted by PDR2 (Fig. 5c and Supplementary Fig. 54).

Building on the idea that distance is a key parameter used to dictate reaction outcome, the effect of increasing the distance between substrate and the iron center in wild-type TsaM was tested. Here, an M230G/T232G variant of TsaM was made and purified (Supplementary Fig. 51). This variant was incubated with VanB and *p*-toluenesulfonate or 4-methylbenzoate. Surprisingly, these reactions each yield two formylated products (**57** and **58**), as well as the expected hydroxymethyl-containing products (**3** and **4**, Fig. 5d, and Supplementary Fig. 63). As described for the sequential monooxygenation reaction on benzenesulfonate and benzoate, a schematic diagram of the reaction progress under multiple turnover conditions reveals an initial appearance of monooxygenated products, consumption of these species, and formation of 4-formyl benzenesulfonate and 4-formyl benzoate (Fig. 5d and Supplementary Fig. 63). These results suggest that lengthening, rather than decreasing, the distance between the substrate and iron center in the active site, permits TsaM, like CAO, to catalyze sequential monooxygenation reactions on the methyl group of its substrate (Figs. 1d and 5d). Once more, using the uncoupling assay, it was revealed that this TsaM variant and substrate combination did not result in any appreciable increase in $H_2O_2$ formation relative to that observed with the wild-type enzyme (Supplementary Fig. 62).

## Rational tuning of Rieske oxygenase reactivity

Additional experiments to determine whether our identified rational strategy for changing a monooxygenase into a dioxygenase could be extended to other Rieske oxygenases were also performed (Fig. 6). For this work, both wild-type VanA and a V232F VanA variant were recombinantly expressed and purified (Supplementary Table 2 and Supplementary Figs. 64 and 65). The motivation for creation of the V232F variant came from a sequence alignment and AlphaFold[52,53] model of VanA, which revealed Val232 is equivalent to TsaM and DdmC residues Thr232 and Ile232, respectively (Supplementary Fig 64). In these experiments, as previously determined[48], wild-type VanA was shown to transform vanillate and 3-methylbenzoate (**15**) into the expected 3,4-dihydroxybenzoate product (**40**) and into a monooxygenated product (**16**), respectively (Fig. 6a and Supplementary Fig. 66). The V232F VanA variant, on the other hand, transforms benzoate into a dioxygenated product that again has the same mass, retention time, and MS/MS fragmentation pattern as the *cis*-dihydrodiol, (3*R*,4*S*)-3,4-dihydroxycyclohexa-1,5-diene-1-carboxylate (Fig. 6a and Supplementary Fig. 67). This result shows that VanA can be predictively tuned to function as a dioxygenase by making a mutation in an equivalent primary sequence position to that chosen for TsaM (Fig. 6a).

An experiment to determine whether the identified hotspot for adjusting reactivity in TsaM and VanA could be extended to a dioxygenase enzyme was also performed. For this endeavor, the active site of PDO was interrogated. Previous structural studies revealed that the active site of PDO anchors phthalate in place using residues Ser182, Arg207, and Arg244 and contains Ile256 at the top of the active site[26] (Fig. 6b). To probe the influence of this residue, which sits in an equivalent position to Met230 of the TsaM model, on the PDO reaction outcome, an I256G variant of PDO was made using site-directed mutagenesis, recombinantly expressed, and purified (Supplementary Table 2 and Supplementary Fig. 68). Using benzoate (**38**) as a substrate, it was determined that unlike wild-type PDO, the I256G variant catalyzes both monooxygenation and dioxygenation transformations that produce molecules that have the same mass, retention time, and

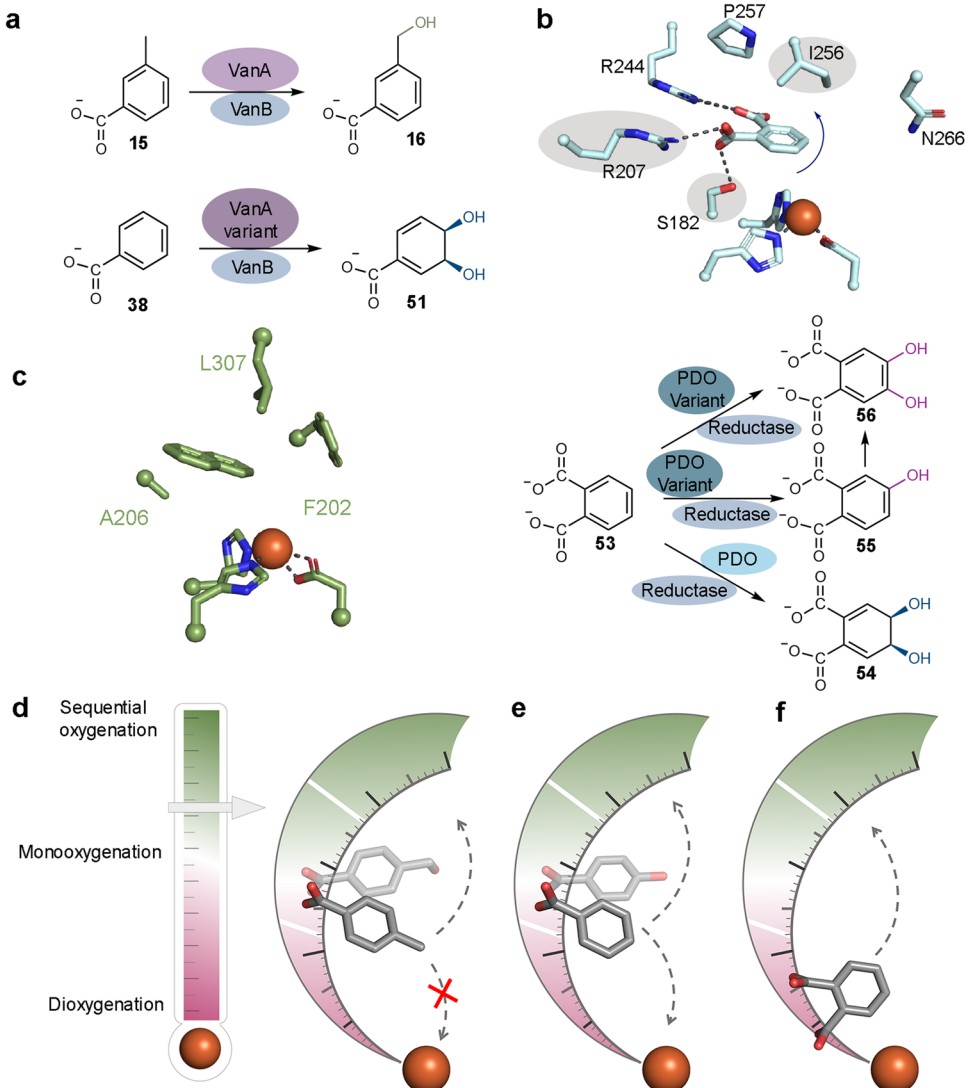

**Fig. 6 | Changing the distance between the substrate and iron center impacts the reaction outcome in other Rieske oxygenases. a** Wild-type vanillate *O*-demethylase (VanA) performs a monooxygenation reaction on 3-methylbenzoate (**15**) to produce 3-(hydroxymethyl)benzoate (**16**) as previously described[48]. A rationally designed VanA variant performs a dioxygenation reaction on benzoate (**38**) to produce a *cis*-dihydrodiol (**51**). **b** Different mutations in the active site of phthalate dioxygenase (PDO, S182I, R207V, and I256G), hypothesized to allow an upward movement of substrate, permit formation of monooxygenated (**55**), sequentially monooxygenated (**56**), and dioxygenated (**54**) species (PDB: 7V25[26]). **c** Residues targeted in naphthalene dioxygenase (NDO) that alter reaction outcome on non-native substrates are highlighted[39] (PDB: 1O7G[9]). **d** In wild-type TsaM, *p*-toluenesulfonate (**1**) and 4-methylbenzoate (**2**) are anchored such that the methyl group is oxygenated. Variants created to move the substrate closer to the iron

center do not support dioxygenation chemistry likely due to clashing. In contrast, variants that permit flexibility in the active site allow CAO-like chemistry (M230G/T232G). **e** Wild-type TsaM catalyzes sequential monooxygenation reactions on benzoate and can be rationally manipulated to exclusively catalyze dioxygenation chemistry (M230F/W, T232F/W, M230W/T232I, M230W/T232I/S257R, and M230W/T232I/S257R/Y269V). A T232I TsaM variant performs both monooxygenation and dioxygenation chemistry. **f** PDO holds its substrate close to the iron center for dioxygenation chemistry. An I256G variant made to promote flexibility permits monooxygenation chemistry, but due to the placement of substrate, dioxygenation is the major chemical outcome. Chemical structures in panels **d–f** were made using the eLBOW tool within the Phenix software suite compiled and configured by SBGrid[94–96].

MS/MS fragmentation pattern as 4-hydroxybenzoate and the *cis*-dihydrodiol (**51**), respectively (Supplementary Figs. 61 and 69). Similarly, unlike wild-type PDO, the I256G variant transforms 4-hydroxybenzoate (**32**) and 4-hydroxyphthalic acid (**55**) into dioxygenated aromatic products 3,4-dihydroxybenzoate and 3,4-dihydroxyphthalate, respectively (Supplementary Fig. 70). Finally, the I256G variant of PDO was shown to catalyze a mixture of monooxygenation and dioxygenation chemistry on phthalate (**53**) (Fig. 6b and Supplementary Fig. 71). It is important to note that although we observe formation of the native dearomatized product, given that 4-hydroxyphthalate is a substrate of I256G PDO, some amount of 3,4-dihydroxyphthalate is also formed in this reaction. Due to the lack of a

commercially available product standard of either dioxygenated product, the individual amount of 3,4-dihydroxyphthalate relative to the *cis*-dihydrodiol (**51**) is unable to be quantified. Encouragingly, however, the presence of the monooxygenated product suggests that the strategic introduction of a small Gly residue into the PDO active site allows for flexibility in the way that substrate binds and can thus account for the observed product mixture (Fig. 6b and Supplementary Fig. 71).

To explore the influence of additional flexibility in the active site of PDO on reaction outcome, two double variants of PDO (R207V/I256G and R207V/I256A) were made (Supplementary Table 2). These double variants targeted Ile256 as well as one of the residues (Arg207) involved in anchoring phthalate close to the iron center (Fig. 6b).

These variants, like the I256G variant, also catalyze monooxygenation chemistry on 4-hydroxybenzoate and 4-hydroxyphthalic acid. These double variants also show altered reactivity with a phthalate substrate (Supplementary Figs. 70 and 71). However, for these double variants, this altered reactivity gives rise to approximately equal amounts of dioxygenated and monooxygenated products (Supplementary Fig. 71). Inspired by this result which suggests that mutation of the residue that anchors phthalate close to the iron center diminishes the native reactivity of PDO, a final triple variant (I256G/S182I/R207V) was made (Supplementary Table 2). Remarkably, this PDO triple variant, for the first time, shows no formation of the native *cis*-dihydrodiol product. Instead, this triple variant exclusively transforms phthalate into a monooxygenated (4-hydroxyphthalic acid) and sequentially monooxygenated (3,4-dihydroxyphthalate) product (Supplementary Fig. 71). In this work we additionally show that the yield of the monooxygenated product formed in these reactions is amplified by substitution of VanB with either PDR1 or PDR2 (Supplementary Fig. 71). Last, for this series of substrates and reductase partners, it was demonstrated that no increased uncoupling occurred relative to the wild-type PDO enzyme system, suggesting that the range of substrates tested, and the introduced mutations, do not profoundly affect the coupling of $O_2$ activation with substrate functionalization (Supplementary Fig. 72).

## Discussion

Enzymes are traditionally considered to exhibit high chemo-, site-, and stereo-selectivity. They capitalize on these properties to catalyze physiologically relevant reactions at rates that meet the needs of a living organism[60]. Today, it is known that many enzymes also accept a broad range of non-physiological substrates and showcase non-native catalytic functions[60–62]. These types of promiscuous behaviors can be advantageous to living organisms because they provide routes for proteins to evolve new functions, and they are also recognized as useful starting points for evolving new or better catalysts[62,63]. The challenge in harnessing these promiscuous functions, however, lies in understanding the intricate structure-function relationships that dictate substrate scope and reaction selectivity in an enzyme of interest. To date, a variety of techniques, including rational and semi-rational design, as well as directed evolution, have been used to pinpoint different design elements that are important for tuning the selectivity and reactivity of a Rieske oxygenase[20,25,40,64–69]. Recent work from our laboratory has demonstrated that the active site and secondary sphere protein residues play a cooperative role in dictating the site-selectivity and substrate scope of two Rieske oxygenases SxtT and GxtA[21,22]. Other important studies have revealed that the oxidation state of the Rieske cluster impacts how the reactive iron-based intermediate of salicylate 5-hydroxylase and benzoate 1,2-dioxygenase interacts with substrate, and thereby dictates whether these enzymes catalyze monooxygenation or dioxygenation chemistry[28,31–33,69,70]. Studies on NDO have additionally revealed that the proximity of the substrate to the iron center facilitates the needed movement of electrons from the Rieske cluster[71]. However, despite the wealth of knowledge that these studies have provided regarding how protein architecture, metallocluster oxidation state, and substrate design influence catalysis, few details exist regarding how a Rieske oxygenase can be predictively tuned to facilitate enantio- and regio-selective reactions that yield a desired product (Fig. 1). Thus, in this work, we sought to identify protein-based elements that can be used to control the outcome of a Rieske oxygenase reaction.

Inspired by the parallels between the three divergent Rieske oxygenase catalyzed reactions illustrated in Fig. 1, in this work, an investigation into whether TsaM could catalyze monooxygenation, dioxygenation, and sequential monooxygenation reactions was undertaken. Through performing a careful evaluation of TsaM activity, the necessary structural features that a substrate must possess to be functionalized by TsaM were identified (Fig. 3). Here, it was determined that TsaM shows the highest levels of activity on substrates that contain a sulfonate or carboxylate functionality at C1 and a methyl group at the *para*-position. Using these substrate design principles and selecting compounds that contain a similar polar or charged functional group to bind in the active site of TsaM, it was demonstrated that fine-tuning of the reaction outcome can be accomplished (Fig. 3a–c). By selecting substrates that contain a carboxylate moiety and longer carbon chains in place of the methyl group, it was demonstrated that TsaM catalyzes both monooxygenation and desaturation chemistry (Fig. 3d). For two of these substrates, 4-ethylbenzoate and 4-isopropylbenzoate, it was confirmed that the benzylic carbon atom is oxygenated, suggesting that the protein architecture biases the positioning of the substrate such that the methyl group, or equivalently positioned benzylic carbon, is similarly oriented with respect to the iron center. On the other hand, by combination of TsaM with substrates that contain methoxy-, methylamino-, or methylthio-substituents in place of the methyl group, TsaM was observed to catalyze oxidative dealkylation reactions (Fig. 3e). As described for substrates *p*-toluenesulfonate, 4-methylbenzoate, *p*-aminotoluene, *p*-nitrotoluene, 4-ethylbenzoate, and 4-isopropylbenzoate, these results again suggest that the protein architecture is biased to position the substrate such that the functional group that is *para* to the sulfonate or carboxylate functionality will be oxygenated.

This trend also holds true in experiments where TsaM is provided with substrates such as benzenesulfonate or benzoate that contain a sulfonate or carboxylate moiety without a methyl group. However, in these cases, the functional group in the *para*-position is a hydrogen atom and functionalization at this position happens only in the first of two steps that lead to formation of a sequentially monooxygenated molecule (Fig. 4 and Supplementary Figs. 36–38). We hypothesize that this altered reaction outcome is correlated with our interrogation of the substrate bound Rieske oxygenase crystal structures (Fig. 5a and Supplementary Fig. 44). This analysis suggests that since TsaM is an annotated monooxygenase, it is built such that the substrate sits further away from the iron center than would be observed in a dioxygenase. As the wild-type enzyme is built to hold the methyl groups of *p*-toluenesulfonate or 4-methylbenzoate close to the iron center, its absence likely allows flexibility in the active site and exacerbates the distance between these entities, meaning that benzenesulfonate and benzoate promote the ability of TsaM to catalyze sequential monooxygenation reactions (Fig. 4). Therefore, TsaM can be thought of as containing a ruler that extends from the iron center into the active site; the distance at which the substrate sits on the ruler controls the reaction outcome (Fig. 6d, e). Substrates that sit at the bottom, middle, and top of the ruler will be converted into dioxygenated, monooxygenated, and sequentially monooxygenated products, respectively. Like other Rieske oxygenases that have been shown to anchor their substrates in the active site using charged functional groups[7,14,16,23,24,26,55,56], we posit that the location on the ruler where the TsaM substrate sits is decided by the C1 moiety (Fig. 6d, e).

Indeed, rationally created TsaM variants that contain Phe, Trp (M230F/W, T232F/W, M230W/T232I, M230W/T232I/S257R, and M230W/T232I/S257R/Y269V), or Ile (T232I) residues at the top of the active site either exclusively produce a *cis*-dihydrodiol in a single step, or a mixture of monooxygenated and dioxygenated species, when provided with a benzoate substrate (Fig. 5b, c). A similar ability of these TsaM variants to perform a dioxygenation reaction on *p*-toluenesulfonate and 4-methylbenzoate is not observed. This lack of activity is presumably due to the construction of the active site, which biases substrate positioning such that the methyl group is always in the correct orientation for functionalization. Attempts to push the substrate closer to the iron center likely result in steric clashing (Fig. 6d). However, rationally created active site variants (M230G/T232G) of TsaM that capitalize on this architectural trait and afford

*p*-toluenesulfonate and 4-methylbenzoate more flexibility in the active site behave like CAO and catalyze sequential monooxygenation reactions to yield a formylated product (Figs. 5d, 6d, Supplementary Figs. 40 and 63).

Consistent with the ruler model, analysis of the PDO active site shows that the C1 and C2 carboxylate substrate functional groups are held close to the iron center[26], presumably supporting native dioxygenation chemistry (Fig. 6b, f). Adding space in the PDO active site through creation of an I256G variant allows room for the substrate to swing up and results in some formation of a monooxygenated product (Fig. 6b, f and Supplementary Fig. 69–71). Additional flexibility afforded by mutations (I256G/S182I/R207V) that disrupt interactions involved in holding phthalate close to the iron center increase the yield of monooxygenated and sequentially monooxygenated products. At the same time, these mutations abolish the native ability of PDO to form a detectable amount of dearomatized product (Fig. 6b and Supplementary Fig. 71). Importantly, these experiments reveal that in the absence of the I256G mutation, no tested PDO variant, reductase, and substrate combination results in monooxygenation chemistry (Supplementary Fig. 71). Therefore, these results, combined with our results on VanA, highlight a hotspot at the top of the active site that can be used to tune the reaction outcome of three divergent Rieske oxygenases (Figs. 5 and 6).

In addition to the observed outcomes of the variant experiments, support for the ruler model as a broader feature of Rieske oxygenases comes from prior structural work on NDO with non-native substrates that support different types of chemistry[38]. Each of these non-native substrates adopt similar binding poses in the active site[38]. However, consistent with our structure-based analysis, in most cases where these non-native substrates are converted into dioxygenated products, the carbon atoms that are functionalized sit closer to the iron center than those of the substrates that are monooxygenated[38]. Additional mutagenesis campaigns performed on NDO with non-native substrates also support the ruler model: targeting the active site in similar spatial locations to those targeted in TsaM promote formation of monooxygenated and dioxygenated product mixtures[39] (Fig. 6c). This result is particularly intriguing as NDO adopts an $\alpha_3\beta_3$ architecture rather than the $\alpha_3$ or $\alpha_3\alpha_3$ architectures of TsaM, VanA, and PDO. As suggested for other metalloenzyme classes, each of these results suggest that regardless of the quaternary architecture, the closest substrate atoms to the iron center will be oxidized[72,73]. Intriguingly, as described above, for NDO, different reaction outcomes are witnessed using non-native substrates or by making subtle alterations in the active site, but these reactions are generally noted to form product mixtures[39,74]. We hypothesize these mixtures are a result of the NDO substrates lacking polar functional groups comparable to the substrates of TsaM, VanA, and PDO. Changes in the active site of NDO afford flexibility but do not permit full control of substrate positioning. Indeed, complete inversion from dioxygenation to monooxygenation chemistry on the same substrate has not been achieved in NDO, likely due to this lack of control. Rational engineering of the reactivity of enzymes like TsaM, on the other hand, can focus on substrate positioning and permit full tuning of reaction outcome (Figs. 5c, d and 6b).

Throughout this work, in several instances, it was shown that some of the tested substrates of TsaM show low turnover numbers and uncouple the $O_2$ activation step from substrate functionalization (*p*-nitrotoluene, phenylacetate, 4-methylphenylacetate, and *p*-isopropyltoluene). Based on the substrate engineering campaign described above, and the putative active site residues involved in substrate recognition, we posit that the presence of the nitro functional group on *p*-nitrotoluene allows for this molecule to adopt a binding pose that resembles that of a native substrate (Fig. 3c, f). However, like phenylacetate, 4-methylphenylacetate, and *p*-isopropyltoluene, and as previously proposed[55,56,75], we hypothesize that an imperfection in the way that *p*-nitrotoluene fits in the native substrate binding site results in

inefficient triggering of subsequent catalytic steps, and therefore leads to uncoupling. On the other hand, many substrates tested simply display decreased total turnover numbers and catalytic efficiencies, without increased $O_2$ uncoupling relative to that observed with *p*-toluenesulfonate and 4-methylbenzoate. As an increasing amount of literature suggests that residues found outside of the Rieske oxygenase active site contribute to substrate positioning[21,22,25,40,48], we hypothesize that this behavior stems from the ability of non-native substrates to assume an ensemble of binding poses in the active site. In essence, providing a substrate that TsaM is not programmed to accept leads to low levels of activity (Fig. 3). For example, there is no obvious trend in activity using *p*-(methylamino)benzoate or *p*-(methylthio) benzoate. Like that previously described for *p*-(methoxy)benzoate[48], each of these molecules have markedly different partial charge distributions relative to *p*-toluenesulfonate and 4-methylbenzoate (Supplementary Fig. 34). We suggest that these different chemical attributes deleteriously impact the ability of this suite of substrates to interact with TsaM and be correctly positioned in the active site (Fig. 3e). We similarly attribute the decreased efficiency of the other investigated TsaM-catalyzed monooxygenation reactions to an inability of TsaM to productively bind substrates that differ with respect to the positions of their methyl group (**15** and **17**) or the bulk of the functional group at the *para*-position (**19, 25, 28, 29,** and **30**), in a manner that supports the needed downstream processes for catalysis (Table 1 and Fig. 3). The former assertion is supported by recent data from our laboratory that shows changes to the active site, substrate entrance tunnel, and a flexible connecting loop are needed to confer TsaM the ability to oxygenate 2-methylbenzoate or 3-methylbenzoate to a significantly greater extent than the wild-type enzyme[48].

An analogous hypothesis can also be applied to the experiments performed with benzenesulfonate and benzoate substrates, which are both smaller than *p*-toluenesulfonate and 4-methylbenzoate. In fact, in the sequential monooxygenation reaction, it is quite remarkable that the products of the first oxygenation event, 4-hydroxybenzenesulfonate and 4-hydroxybenzoate, are oxygenated in the presence of a large excess of benzenesulfonate and benzoate. We attribute this phenomenon to the size of the monooxygenated compounds which more closely resemble *p*-toluenesulfonate and 4-methylbenzoate and likely supports a near-native binding orientation in the active site (Supplementary Fig. 34). Similarly, it is worth mentioning that relative to benzoate, the partial charge distribution of 4-hydroxybenzoate more closely approximates *p*-(methoxy)benzoate, a molecule that is turned over to a significantly greater extent than benzoate by TsaM (Supplementary Figs. 34 and 38f). This latter point again suggests that a higher proportion of the monooxygenated compounds will adopt productive binding orientations relative to that of benzenesulfonate and benzoate in the TsaM active site. Quite interestingly, the kinetic parameters for each of these different molecules are nearly equivalent (Table 1). The basis of this phenomenon, at this point, is unclear. Typically, this finding would suggest that the rate of $O_2$ activation in the presence of the different substrate options is similar. However, we anticipate that the values measured for this set of substrates are not directly comparable due to the sequential nature of the reaction. Rather, as described above, it is possible that the chemical attributes of these molecules allow for an ensemble of binding poses in the protein active site. The proportion of these molecules that are positioned correctly permit catalysis and immediately produce either 4-hydroxybenzenesulfonate and 4-hydroxybenzoate, which are also substrates of the reaction. These measured kinetic parameters with each these different molecules will be the basis of future investigation. However, we still posit that the monooxygenated compounds are expected to better bind in the active site and support the needed downstream processes for catalysis. Perhaps related to this interpretation, here it is shown that NADH consumption is increased in the presence of 4-hydroxybenzoate and

4-hydroxybenzenesulfonate relative to their non-oxygenated counterparts (Supplementary Fig. 38).

Similar parallels can be made for the tested TsaM variants (M230F/W, T232F/W, M230W/T232I, M230W/T232I/S257R, or M230W/T232I/S257R/Y269V). Here, the active sites are engineered to confer dioxygenation chemistry, but the protein scaffold, as mentioned above, is not optimized for a benzoate substrate. As such, some of the benzoate molecules likely adopt binding orientations that are productive, and others adopt poses that negatively impact the protein behaviors coupled to catalysis (Fig. 5c). In contrast, the non-native sequential monooxygenation and monooxygenation activity of TsaM and PDO variants is generally higher in the presence of the reported native substrates (Figs. 5d, 6b, and Supplementary Figs. 69–71). In particular for the M230G/T232G TsaM variant, it is interesting to note that the polar charge distributions at the $p$-positions of the reported native products, $p$-(hydroxymethyl)benzenesulfonate and 4-(hydroxymethyl)benzoate, resemble the polar charge distributions at the $p$-positions of 4-hydroxybenzenesulfonate and 4-hydroxybenzoate (Supplementary Fig. 34). However, $p$-(hydroxymethyl)benzenesulfonate and 4-(hydroxymethyl)benzoate have added steric bulk conferred by the additional methylene group. This extra bulk is likely the culprit that prevents wild-type TsaM from catalyzing a CAO-like second reaction on these molecules, but it is intriguing that the introduction of two active site Gly residues, a presumable extension of the active site ruler, does permit sequential monooxygenation reactions on $p$-toluenesulfonate and 4-methylbenzoate (Fig. 5d and Supplementary Fig. 40). Collectively, these results suggest that combination of a native substrate with an altered enzyme active site permits better control over downstream catalytic behaviors, including $O_2$ binding, electron transfer, and a productive interaction with the reductase[29,30,76]. Such an interaction has been previously shown to induce conformational changes in the Rieske oxygenase, and therefore, could presumably also influence reaction outcome[29,30]. Indeed, whereas clear differences in the activity of TsaM and PDO with native and non-native reductases were observed in this work, the basis of these differences remains to be determined.

Future work aimed at understanding the origin of these differences and investigating the value of combining the strategies discovered here with other previously described methods for tuning outside of the active site residues[21,22,40,77], is paramount. Furthermore, it is important to investigate whether the lower observed activity of engineered Rieske oxygenases could be due to a disruption in the relationship with the noted downstream catalytic behaviors or attributed to a phenomenon described for other designed enzymes wherein changes in substrate binding disturb the protein rapid dynamics that are coupled to catalysis[78–80]. Overall, the results presented here suggest that reaction outcome in TsaM, VanA, and PDO is governed by a ruler in the active site that can be rationally manipulated to change catalytic outcome (Fig. 6d–f). Therefore, this work provides predictive power to rationally tune other members of the Rieske oxygenase family to catalyze a promiscuous reaction of interest. These results further provide important considerations for design and synthesis of small molecule catalysts and artificial metalloproteins.

## Methods

### Site directed mutagenesis of TsaM, PDO, and VanA
For each of the variants made in this work a Bio-Rad C1000 Thermal Cycler and the Agilent QuickChange Lightning Site-Directed Mutagenesis Kit were employed. In brief, mutagenesis reactions were prepared on a 50 μL reaction scale and required the addition of 200 ng of the needed plasmid (pET-28a(+)-TEV-*tsaM*, pMCSG9-*pdo*, or pMCSG9-*vanA*) and 125 ng of each oligonucleotide primer (synthesized by IDT). Each DNA and protein sequence is provided in a Supplementary Data 1 File and each primer sequence is supplied in Supplementary Table 2. Following PCR, digestion was accomplished using 2 μL of DpnI, and the reaction mixtures were transformed into either XL10-Gold or JM109 (Agilent) competent cells. Successful mutations were individually verified by Sanger sequencing (Genewiz).

### Protein expression conditions for TsaM and TsaM variants
TsaM was expressed and purified as previously described[48]. In brief, a pET-28a(+)-TEV-*tsaM* plasmid was transformed into *Escherichia coli* C41(DE3) cells (Novagen) and plated on lysogeny broth (LB)-kanamycin-agar plates. A single colony was used to inoculate a 5 mL starter culture of LB containing 50 μg/mL kanamycin and this starter culture was used to inoculate 1 L of LB-kanamycin in a 2 L flask. Four of these 1 L cultures were grown in a constant temperature of 37 °C until the $OD_{600}$ reached 0.6-0.7. At this point, the temperature was cooled to 20 °C and 0.1 mM isopropyl β-D-1-thiogalactopyranoside (IPTG), 0.2 mg/mL ferric ammonium citrate, and 0.4 mg/mL ferrous ammonium sulfate hexahydrate were added to the cultures. Each of the cultures were incubated for an additional 18 h with shaking (200 rpm) prior to harvesting. For wild-type TsaM, the typical wet mass of a pellet from a 1 L culture was approximately 5 g.

### Purification protocol for TsaM and TsaM variants
As previously described[48], the cells containing overexpressed His-tagged TsaM protein were harvested by centrifugation. This cell pellet was resuspended in lysis buffer (50 mM Tris-HCl (pH 7.2), 250 mM NaCl). Cells were lysed at 4 °C using a Fisherbrand Model 120 Sonic Dismembrator in pulse mode. Following completion of a 5 min program that cycles between 10 s sonication and 20 s rest, the lysed cells were clarified by centrifugation. Following this step, the supernatant was loaded onto a 5 mL HisTrap column (Cytiva), which was washed with Buffer A (50 mM Tris-HCl (pH 7.2), 250 mM NaCl, and 5 mM imidazole) and eluted with Buffer B (50 mM Tris-HCl (pH 7.2), 250 mM NaCl, 200 mM imidazole, and 5% glycerol). The elution fractions were concentrated and loaded onto a HiPrep 16/60 Sephacryl S200-HR (Cytiva) gel filtration column. This column was pre-equilibrated and run with Buffer C (50 mM HEPES, pH 8.0, 200 mM NaCl, and 5% glycerol). The TsaM protein that eluted from this column in a trimeric state (~120 kDa), determined based on a gel-filtration standard (Bio-Rad), was collected and concentrated to 10 mg/mL. Protein aliquots (50 μL) were then prepped for storage in a −80 °C freezer.

### Protein expression conditions for VanB, TsaB, PDR1 and PDR2
The methods for expressing and purifying VanB and TsaB were previously described and followed here[48,81]. In brief, the constructs for expressing these proteins consist of pMCSG7-*vanB* and pET28a(+)-TEV-*tsaB*. For PDR1, and PDR2, codon optimized genes encoding PDR1 and PDR2 were synthesized and cloned into a pET-28a(+)-TEV plasmid (Genscript). The pET28a(+)-TEV-*pdr1* or pET28a(+)-TEV-*pdr2* plasmids were transformed into C41(DE3) *E. coli* cells (Novagen). As described for TsaM, single colonies were used to grow starter cultures at 37 °C overnight for VanB, TsaB, PDR1, and PDR2. The starter cultures were used to inoculate 1 L of LB containing the appropriate antibiotic. For TsaB, terrific broth (TB) was used in lieu of LB. The large cultures were incubated at 37 °C until the $OD_{600}$ reached 0.6−0.8. Cells were induced with 0.1 mM IPTG and incubated for an additional 18 h with shaking until harvesting.

### Purification protocol for the reductase proteins VanB, TsaB, PDR1 and PDR2
The cell pellets from two 1 L cultures were resuspended in 50 mL of Buffer A (50 mM Tris-HCl (pH 7.2), 200 mM NaCl, and 20 mM imidazole) that also contained 100 μM flavin adenine dinucleotide (FAD). The mixtures were lysed by sonication and centrifuged using the protocol described above. The supernatants were loaded onto a 5 mL HisTrap column (Cytiva) and a HiPrep 16/60 Sephacryl S200-HR (Cytiva) gel filtration column using the protocols described above.

For VanB and TsaB, the buffers used for TsaM were employed here. In contrast, for PDR1 and PDR2, Buffer B (20 mM Tris-HCl (pH 7.2), 200 mM NaCl, 20 mM imidazole) and Buffer C (20 mM Tris-HCl (pH 7.2), 200 mM NaCl, 200 mM imidazole) were used for washing and eluting the protein from the HisTrap column. The purified proteins were concentrated to 200 μM and flash frozen for long-term storage at −80 °C. FAD incorporation into the reductase proteins was calculated using the UV-Vis absorbance at 450 nm and an extinction coefficient of 11,300 $M^{-1}$ $cm^{-1}$.

### Protein expression conditions for phthalate dioxygenase (PDO)

A codon optimized gene encoding PDO was synthesized and cloned into a pMCSG9 plasmid containing an MBP tag and TEV cleavage site (Genscript). pMCSG9 plasmids containing the *pdo* gene were transformed into C41(DE3) *E. coli* cells. A single colony was used to inoculate 8 mL LB containing 100 μg/mL ampicillin and incubated at 37 °C, 200 rpm overnight. The overnight culture was used to inoculate a 1 L of LB culture in a 2.8 L flask containing 100 μg/mL ampicillin. Cultures were incubated at 37 °C and 200 rpm until an $OD_{600}$ of 0.6–0.8 was achieved. Flasks were chilled at 20 °C for 1 h before induction by addition of 0.1 mM IPTG, 0.2 mg/mL ferric ammonium citrate, and 0.4 mg/mL ferrous ammonium sulfate hexahydrate. Cultures were incubated overnight at 20 °C and 200 rpm overnight (-20 h) before harvesting. The typical wet mass of a pellet from a 1 L culture was 5 g.

### Protein purification protocol for PDO

To purify the wild-type PDO for enzymatic assays, approximately 10 g of cell pellet from 2 L of cell culture was resuspended in 60 mL of Buffer A (20 mM TrisHCl pH 7.4, 200 mM NaCl, and 1 mM DTT). Cells were then lysed and centrifuged as described for TsaM. Following centrifugation, the supernatant was loaded at 2.5 mL/min onto an MBP-Trap column (Cytiva) that was preequilibrated with Buffer A. The column was washed with 6 column volumes of Buffer A at 2.5 mL/min after which the MBP-tagged PDO was eluted with 100% Buffer B (20 mM TrisHCl pH 7.4, 200 mM NaCl, 1 mM DTT, and 10 mM maltose) over 6 column volumes at 2.5 mL/min. The elution fractions were pooled and diluted to 30 mL with the addition of 2 mg of tobacco etch virus (TEV) protease, and this mixture was dialyzed in a 10,000 MWCO snakeskin dialysis tubing overnight in Buffer A. Following this step, the tag-cleaved protein was loaded onto a pre-equilibrated 1 mL His-Trap column (Cytiva) at 1.5 mL/min. The flowthrough fractions containing cleaved PDO were collected and exchanged into storage Buffer C (50 mM HEPES pH 8.0, 200 mM NaCl, and 5% glycerol), and concentrated to 200 μM. The final purified red-brownish protein was flash frozen by liquid nitrogen for storage at −80 °C.

### Protein expression conditions for vanillate *O*-demethylase (VanA)

VanA was expressed and purified as previously described[48]. In brief, single C41(DE3) *E. coli* cell colonies that contained the VanA-encoding plasmid were used to grow 5 mL starter cultures of LB containing 100 μg/mL ampicillin. These small cultures were grown overnight at 37 °C and 200 rpm and used to inoculate 4 larger 1 L LB cultures prepared in 2 L flasks. The 1 L cultures were grown at 37 °C and 200 rpm until the $OD_{600} = 0.8$. Cultures were then induced by addition of 0.1 mM IPTG, 0.2 mg/mL ferric ammonium citrate, and 0.4 mg/mL ferrous ammonium sulfate hexahydrate. Cultures were incubated at 20 °C and 200 rpm overnight prior to harvesting.

### Protein purification protocol for VanA

Using previously described methods[48], wild-type VanA and variants of VanA were purified for activity assays. In brief, approximately 10 g of cell pellet was resuspended in lysis buffer (20 mM Tris-HCl (pH 7.4), 200 mM NaCl, and 1 mM DTT). Cells were lysed and centrifuged using the same protocol described for TsaM. The supernatant sample after centrifugation was loaded onto a Bio-Rad FPLC system fitted with a

5 mL MBPTrap column (Cytiva). The column and subsequent MBP-tag-cleavage protocols described above for PDO were followed. The resultant tag-free VanA was exchanged into storage Buffer C (50 mM HEPES pH 8.0, 200 mM NaCl, and 5% glycerol), concentrated to 200–250 μM using a 30 kDa MWCO centrifugal filter, and stored at −80 °C.

### Iron quantification for isolated TsaM, PDO, and VanA

The iron content of isolated TsaM, PDO, and VanA was determined following a published procedure that employs a spectrophotometric reagent for iron analysis ($\varepsilon_{593} = 34,500$ $M^{-1}$ $cm^{-1}$)[48,82]. The number of iron ions incorporated into isolated TsaM, PDO, and VanA was determined to be approximately 3 per monomer.

### Circular dichroism (CD) experiments

To collect CD spectra for both wild-type TsaM and variants TsaM, we capitalized on the CD spectra collection methods that were used in our previous work[21,48]. These methods use 350 μL samples of 5–10 μM wild-type TsaM and TsaM variants, a 10 mm quartz cuvette (Hellma), and a Jasco J-1500 CD spectrometer.

### Hydrogen peroxide ($H_2O_2$) shunt experiments performed with TsaM

The $H_2O_2$ shunt reactions that were conducted included 200 μM of substrate, 100 μM ferrous ammonium sulfate hexahydrate, and 40 μM TsaM. Once combined, either 200 μM $H_2O_2$ or 200 μM $H_2^{18}O_2$ was added to initiate chemistry. The reactions were subsequently incubated at 30 °C for 3 h and quenched with 100 μL of an acetonitrile mixture that contained an internal standard. Quenched reactions were centrifuged at $17,000 \times g$ for 15 min and 50 μL of the supernatant was diluted with 100 μL of acetonitrile. The reaction mixture was analyzed using quadrupole-time of flight (qTOF) LC-MS and the methods described below.

### Initial enzymatic assays to probe the needed components for TsaM activity

The identity of all compounds used in the activity assays is included in Supplementary Table 1. Additional relevant information regarding the supplier of these molecules is provided in a Supplementary Data 1 File. Initial enzymatic assay reactions were prepared to contain 4 mM substrate, 500 μM NADH, 100 μM ferrous ammonium sulfate hexahydrate, 10 μM TsaM, and 40 μM of VanB or TsaB. For these reactions, TsaM was added last. Once combined, the reactions were mixed and incubated at 30 °C for 3 h. After incubation, reactions were quenched by the addition of 100 μL acetonitrile containing the appropriate mass spectrometry internal standard. The quenched reaction mixtures were then centrifuged at $17,000 \times g$ for 15 min. Following centrifugation, 50 μL of the supernatant was run on liquid chromatography mass spectrometry (LC-MS) as described below. These initial reactions confirmed, as previously described[48], that VanB was a more efficient reductase for TsaM than TsaB. However, to ensure the optimal reaction conditions, additional tests were performed to determine the ideal reductase:TsaM ratio. All data was housed in Microsoft Excel.

These reductase tests were performed on a 50 μL scale. Each reaction contained 2 mM substrate (*p*-toluenesulfonate or 4-methylbenzoate), 500 μM NADH, 100 μM ferrous ammonium sulfate hexahydrate, 5 μM TsaM, and either TsaB or VanB. Reductase proteins were initially added in different concentrations relative to TsaM (i.e., in a ratio of 2:1, 3:1, 4:1, and 5:1). The reaction mixtures were incubated at 30 °C for 3 h and quenched as described above. The optimal ratio of reductase:TsaM was determined to be 4:1.

Next, a study was undertaken to determine the optimal reaction time for product formation. In this study, a 50 μL reaction mixture containing 2 mM substrate, 500 μM NADH, 100 μM ferrous ammonium sulfate hexahydrate, 5 μM TsaM, and 20 μM TsaB or VanB was

incubated at 30 °C and quenched at 1, 2, 3, 4, and 6 h time points. These reactions were quenched, centrifuged, and analyzed using the LC-MS methods described below. Here, it was determined that the maximum amount of product was formed using a 3 h incubation.

## Total turnover number (TTN) determination for TsaM

To generate plots of total turnover numbers, 50 μL reactions were prepared containing 2.5 μM TsaM, 10 μM VanB, 1 mM of each tested substrate [p-toluenesulfonate (1), 4-methylbenzoate (3), 3-methylbenzoate (15), 2-methylbenzoate (17), p-aminotoluene (5), p-nitrotoluene (6), p-isopropyltoluene (7), toluene (11), p-chlorotoluene (13), 4-ethylbenzoate (19), 4-isopropylbenzoate (25), 4-propylbenzoate (28), 4-butylbenzoate (29), 4-pentylbenzoate (30), p-(methoxy)benzoate (31), p-(methylamino)benzoate (33), p-(methylthio)benzoate (34), benzenesulfonate (37), 4-hydroxybenzenesulfonate (41), benzoate (38), 4-hydroxybenzoate (32), aniline (42), phenol (43), N-phenylacetamide (44), phenyl acetate (45), 4-methylphenyl acetate (46), benzoylformate (47), 4-methylbenzoylformate (48)], 1 mM NADH and 100 μM ferrous ammonium sulfate hexahydrate. These reactions were mixed and incubated at 30 °C for 3 h before quenching using a 100 μL solution of acetonitrile that contained our mass spectrometry internal standard acetaminophen or 3,5-dihydroxyacetophenone as previously described[48]. The latter internal standard was used only for 33, 44, and their corresponding products due to the identical mass of acetaminophen and the singly oxygenated products of 33 and 44. The quenched reaction mixtures were centrifuged at $17,000 \times g$ for 15 min. Following centrifugation, 50 μL of the supernatant was run on LC-MS and analyzed as described below. Of note, TTN assays were also tested with 4 mM substrate to ensure that the substrate concentration was not a limiting factor in the reaction.

In most cases, the total amount of product generated in these reactions was determined using standard curves of the reaction products that were constructed using commercially available product standards and previously described methods[48]. The ratio obtained from the enzymatic assays was used to calculate the concentration of product formed in the reactions. For the substrates 28, 29, and 30, the products generated were instead calculated based on substrate consumption and employment of a standard curve of the substrate.

TTNs were calculated as previously described[48]. All of the standard curves that were used in this work were constructed from duplicate standard runs. Enzymatic reactions were prepared and analyzed in triplicate. Standard curves of products materials are shown in Supplementary Figs. 73 and 74. In all cases, since each monomeric unit of a Rieske oxygenase contains one active site, the TTN was calculated using the concentration of a single protomer. We note that all substrate and product standards used in this study were prepared as previously described[48] in dimethyl sulfoxide (DMSO, analytical grade). All data was housed in Microsoft Excel. All activity assays and MS data were visualized using ProFit and GraphPad Prism 9 software.

## Jones oxidation on TsaM reaction products

Enzymatic reactions were carried out as described above for TTN determination with minor adjustments. Namely, in these assays, the enzymatic assay reaction mixture components were added in four-fold excess to generate a total reaction volume of 200 μL. The increased reaction volume was imperative for generation of more product. Enzymatic reactions were quenched with a 200 μL mixture of acetonitrile and the mass spectrometry internal standard acetaminophen. Following 3 h incubation at 30 °C, chromium trioxide was added. A 100 mM stock solution of chromium trioxide was prepared in in concentrated sulfuric acid. A 100 μL aliquot of chromium trioxide solution was then combined with 300 μL of the quenched enzymatic reaction mixture, 200 μL acetone, 200 μL water, and incubated at room temperature for 3 h, with constant stirring. The reaction mixture was then diluted 1:1 with sterile filtered acetonitrile containing acetaminophen

as an internal standard. The reaction mixture was further analyzed using the LC-MS methods described below.

## Synthesis of the dioxygenated product of N-phenylacetamide (44)

Due to the difficulty of purchasing the dioxygenated product of 44, the compound standard was synthesized by using an amine protecting group. In more detail, 500 μM of either 4-aminophenol or 3,4-dihydroxyaniline dissolved in methanol was mixed with 300 μL acetic anhydride. This solution was incubated at room temperature under constant stirring overnight. The next day, 50 μL of this reaction mixture was diluted with 100 μL acetonitrile containing 3,5-dihydroxyacetophenone as an internal standard, and it was further analyzed using LC-MS methods described below.

## Kinetic analysis of TsaM with tested substrates

To measure the apparent steady-state kinetic parameters of TsaM, an approach similar to that previously described was undertaken[48]. First, the linear range for product formation was investigated with each tested substrate (Supplementary Table 1). Here, all enzymatic reactions were performed on a 50 μL scale with 400 μM substrate, 1 mM NADH, and 100 μM ferrous ammonium sulfate hexahydrate. In addition, three different TsaM concentrations at 2, 5, and 10 μM were used (the concentration used was chosen based on the measurability of the product signal). The concentration of VanB included in the assays was always four times of TsaM concentration used. These reactions were quenched at 2, 5, 10, or 40 min with the internal standard-acetonitrile mixture in 1:1 ratio.

After identification of the linear range for product formation using each substrate, the apparent steady-state kinetic parameters of TsaM with different substrates were determined using enzymatic reactions conducted on the 50 μL scale. Each reaction included 500 μM NADH, 100 μM ferrous ammonium sulfate hexahydrate, and various amounts of the substrate under investigation. As described above, the concentration of TsaM (and VanB) differed based on what was determined in the initial trials. All enzymatic assays were initiated by the addition of TsaM. Reactions were quenched at a time point in the linear range using 50 μL of the acetonitrile-internal standard mixture. Quenched reactions were centrifuged at $17,000 \times g$ for 15 min. Finally, 1–2.5 μL of each sample was analyzed using the LC-MS protocols described below. The generated product in each reaction was quantified using standard curves as described above. The data was plotted and the kinetic parameters were extracted using GraphPad Prism 9 software. All kinetic reactions were performed in duplicate. In all cases, since each monomeric unit of a Rieske oxygenase contains one active site, the $k_{cat}$ was calculated using the concentration of a single protomer. All data was housed in Microsoft Excel.

## PDO and VanA enzymatic reactions

Reactions that used PDO or its variants were performed similarly to those described for TsaM. Fresh protein aliquots were used for each enzymatic assay, and 50 μL reaction mixtures were prepared to contain 50 μM PDO, 200 μM VanB/PDR1/PDR2, 500 μM NADH, and 100 μM ferrous ammonium sulfate hexahydrate. The enzymatic reactions were initiated by the addition of 2 mM substrate. This reaction mixture was then incubated at 30 °C for 3 h and followed by the quench procedure described above. A 50 μL aliquot of the centrifuged supernatant was subsequently taken to analyzed using LC-MS and the methods described below. All data was housed in Microsoft Excel. All activity assays and MS data were visualized using ProFit and GraphPad Prism 9 software.

## O₂ uncoupling assay

The amount of $O_2$ uncoupling in the wild-type TsaM and TsaM variant reactions was inspired by previous work[55,56] and employed previously

described methods[48]. To accomplish these measurements and obtain an estimate of the extent of $O_2$ uncoupling in our reactions, we capitalized on the availability of an Invitrogen Amplex Red hydrogen peroxide/peroxidase assay kit. The detection of $H_2O_2$ produced in the reactions was determined by incubation of 50 μL enzymatic reactions (2 mM substrate, 500 μM NADH, 100 μM ferrous ammonium sulfate hexahydrate, 5 μM oxygenase, and 20 μM reductase) with 50 μL of Amplex Red reagent/horse radish peroxidase solution. The experiment was performed in a 96-well plate, the absorbance at 560 nm was measured every 5 min for a total of 190 min using a BioTek Epoch2 microplate reader, and the reactions were initiated by the addition of wild-type TsaM or TsaM variant. Of note, this experiment exclusively provides a lower estimate of the unproductively activated $O_2$ that is lost as $H_2O_2$. All data was housed in Microsoft Excel.

### Intact mass spectrometry experiments

To probe for a reactive oxygen species mediated protein-based modification in the enzymatic assays, LC-MS analysis was performed using an Agilent G6545A qTOF mass spectrometer equipped with a dual AJS ESI source and an Agilent 1290 Infinity series diode array detector, autosampler, and binary pump. For these experiments, an Aeris WIDEPORE C4 column (2.1 × 50 mm, 3.6 μm, 200 Å) (Phenomenex) was employed. Each tested sample contained 10 μM oxygenase, 40 μM reductase, 2 mM substrate, 500 μM NADH, and 100 μM ferrous ammonium sulfate hexahydrate and was incubated for 3 h prior to analysis. The LC-MS solvents used were solvent A (95% water, 5% acetonitrile, and 0.1% formic acid) and solvent B (95% acetonitrile, 5% water, and 0.1% formic acid). The seven-minute chromatographic method used for these experiments employed a gradient that ran from 5% to 95% Solvent B with a flow rate of 0.3 mL/min. Expected protein molecular weights were calculated using Agilent BioConfirm software.

### LC-MS and MS/MS analysis of enzymatic reactions

LC-MS analysis was performed as previously described[48] using the system described above. For most of the substrates and products, the LC-MS solvents used were solvent A (5% acetonitrile, 95% water, and 20 mM ammonium acetate pH 5.5) and solvent B (95% acetonitrile, 5% water and 20 mM ammonium acetate pH 5.5). The instrument was run in negative ion mode. For *p*-aminotoluene (**5**), *p*-nitrotoluene (**6**), *p*-isopropyltoluene (**7**), toluene (**11**), *p*-chlorotolune (**13**), aniline (**42**), phenol (**43**), and their products, solvent C (5% acetonitrile, 95% water, and 0.1% formic acid) and solvent D (95% acetonitrile, 5% water, and 0.1% formic acid) were used. The data collection for these compounds were performed in positive ion mode.

An Agilent ZORBAX Rapid Resolution HT 3.5 μm, 4.6 ×75 mm SB-CN liquid column was used for separation and analysis of *p*-toluenesulfonate (**1**) and 4-methylbenzoate (**2**) as previously described[48]. Here, this column was also used for analysis of reactions that contained *p*-aminotoluene (**5**), *p*-isopropyltoluene (**7**), 3-methylbenzoate (**15**), 2-methylbenzoate (**17**), 4-ethylbenzoate (**19**), 4-isopropylbenzoate (**25**), 4-propylbenzoate (**28**), 4-butylbenzoate (**29**), 4-pentylbenzoate (**30**), *p*-(methoxy)benzoate (**31**), *p*-(methylamino)benzoate (**33**), *p*-(methylthio)benzoate (**34**), benzenesulfonate (**37**), benzoate (**38**), aniline (**42**), *N*-phenylacetamide (**44**), phenyl acetate (**45**), 4-methylphenyl acetate (**46**), benzoylformate (**47**), 4-methylbenzoylformate (**48**), phthalate (**53**) and their corresponding oxidized products. The chromatographic method used for these substrates and this Agilent SB-CN column typically used 10% Solvent B from 0 to 1 min (to waste) and a gradient that ran from 10% to 95% Solvent B from 1.0 to 4.0 min, which was followed by a 1–1.5 min isocratic flow at 95% Solvent B. In most cases, the mass spectrometry was run in negative ion mode. When **5** and **42** were used as substrates, the instrument was run in positive ion mode as described above.

An alternative column, the Agilent ZORBAX Rapid Resolution HT 1.8 μm, 2.1 ×50 mm SB-Aq liquid column was used for separation and analysis of *p*-nitrotoluene (**6**), *p*-chlorotoluene (**13**), and their corresponding oxygenated products. The method for using this Agilent SB-Aq column to analyze the reactions containing these substrates was as follows: 5% Solvent B from 0 to 1 min (to waste), a gradient that ran from 10% to 60% Solvent B for from 1 to 2 min, and a second gradient from 60% to 95% Solvent B from 2 to 3 min, followed by a 1.5 min isocratic flow of 95% Solvent B (to MS). The column was then re-equilibrated with 10% Solvent B for another 1 min (to waste). The whole run was performed at a constant flow rate of 0.4 mL/min. Based on measurability of the product signal, 1–5.0 μL volume of sample was injected into the LC-MS system. Due to the difficulty associated with the ionization of *p*-chlorotoluene (**13**) and its oxygenated product, the UV-absorbance signal at 260 nm was used as a proxy to detect substrate consumption and product generation. This method capitalized on the in line diode array detector of the qTOF. The mass spectrometry was run in positive ion mode as described above.

Finally, toluene (**11**) and phenol (**43**) and their oxygenated products were separated and analyzed using a ThermoScientific Syncronis 1.7 μm, 2.1 ×50 mm aQ C18 column. The method for using this Fisher C18 column was as follows: 5% Solvent B from 0 to 1 min (to waste), a gradient that ran from 10% to 60% Solvent B from 1 to 2 min, and a second gradient from 60% to 95% Solvent B from 2 to 3 min, followed by a 1.5 min isocratic flow of 95% Solvent B (to MS). The column was then re-equilibrated with 10% Solvent B for another 1 min (to waste). Due to the difficulty associated with the ionization of toluene (**11**), the UV-absorbance signal at 260 nm was used as a proxy to detect substrate consumption. This method capitalized on the diode array detector of the qTOF. The mass spectrometry was run in positive ion mode as described above.

Targeted MS/MS experiments were performed with the same mass spectrometry equipment, column and the chromatographic methods described above to get the desired fragmentation patterns. MS/MS methods were adjusted to the corresponding mass of the target compound. Collision energies were set to 5, 10, and 20 eV for all compounds. All data was analyzed using Agilent MassHunter software. All activity assays and MS data were visualized using ProFit and GraphPad Prism 9 software.

### Density functional theory calculations

Initial geometries for substrates, including *p*-toluenesulfonate (**1**), 4-methylbenzoate (**2**), 4-(hydroxymethyl)benzenesulfonate (**3**), 4-(hydroxymethyl)benzoate (**4**), 4-hydroxybenzoate (**32**), benzoate (**38**), *p*-(methoxy)benzoate (**31**), *p*-(methylamino)benzoate (**33**), *p*-(methylthio)benzoate (**34**), benzenesulfonate (**37**), and 4-hydroxybenzenesulfonate (**41**) were built and pre-minimized in the Avogadro software package[83] and computational methodology followed that as previously described[48]. Geometry optimizations followed by single point calculations were carried out using the ORCA software package, version 4.2.1[84,85]. The B3LYP functional[86,87] with Becke–Johnson damping dispersion correction, D3BJ in ORCA syntax[88,89], was used for all calculations. Each calculation used the Alrichs TZVPP basis set[90], with the auxiliary def2-J basis set[91], on every atom. Slow convergence and tight self-consistent field requirements were employed along with an integration grid with an increased number of points (Grid6 in ORCA syntax). The RIJCOSX approximation was used to increase calculation speed[92,93]. Electrostatic potential surfaces were generated from ORCA output and subsequently visualized using Avogadro[83].

### Reporting summary

Further information on research design is available in the Nature Portfolio Reporting Summary linked to this article.

## Data availability

Source data are provided with this paper. The underlying data generated for Figs. 3–5 and Supplementary Figs. 6–10, 15, 17, 18, 22, 26, 31–33, 35–38, 43, 52-53, 62-63, 68-74 in this study are provided in the Source Data files. Other data are available in the Supplementary Information. The crystal structure data used in this study are available in the Protein Data Bank under accession codes 3GL2, 7V25, 1O7G, 7SZH, 7SZE, 1Z03, 6ICN, 6Y9D, 1ULJ, 6LL0, 2BMQ, and 3EN1.

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

## Acknowledgements

The work in this publication was supported by the National Institute of General Medical Sciences of the National Institutes of Health under Award Number R35 GM138271 (J.B.R.). J.B.R. is a Searle Scholar. The contents of this publication are solely the responsibility of the authors and do not necessarily represent the official views of NIGMS or NIH. The VanA construct used in this was produced by the Joint Genome Institute. This work (proposal: https://doi.org/10.46936/10.25585/60000495) conducted by the U.S. Department of Energy Joint Genome Institute, a DOE Office of Science User Facility, is supported by the Office of Science of the U.S. Department of Energy operated under Contract No. DE-AC02-05CH11231. We thank Johnny Mendoza for insightful early insights and discussions regarding the chemistry of TsaM, and the University of Michigan Natural Product Discovery Core for use of the MassHunter software.

## Author contributions

J.T., J.L., M.K., D.G.B., P.H.D., and J.B.R. contributed to the design of the experiments and wrote the manuscript. J.T. prepared materials, performed all biochemical assays, and obtained all LC-MS data. J.L. performed structural analysis. M.K. assisted in product identification and needed synthesis. D.G.B. made the AlphaFold model to assist with rational design efforts, and P.H.D. performed substrate calculations.

## Competing interests

The authors declare no competing interests.
