## [Peer Review File · Nature Communications]

REVIEWER COMMENTS

Reviewer #1 (Remarks to the Author):

Tian et al. describes the Rieske oxygenase reactivity of TsaM from *Comamonas testosteroni*. They found TsaM could catalyze a variety of different reactions, including monooxygenation, and sequential monooxygenation. They considered the distance between substrate molecules and iron atoms are the key for the reactions and tune the reaction activity of TsaM/VanA via site-directed mutagenesis. Some interesting findings were observed, including different oxygenation reactions in different enzymatic systems and the tuning of VanA reactivity via mutagenesis. However, I have some concerns.

1. Data in figure 4d and 5d is not convincing. a) Only relative intensity was shown, not supporting the products generated from a sequential oxygenation, and b) the reaction time was too long to exclude non-enzymatic oxygenation.
2. A model is suggested to show why the reactivity tuning could be achieved. For now, only the distance between substrate molecules and iron atoms were mentioned.
3. The manuscript could be improved after a detail check, especially for Introduction.

Abstract

Please clearly show your contribution. More technologic details should be helpful.

Introduction

Para. 1,

Sentence starting with "For example", references should be cited.

Sentence starting with 'Collectively', the Rieske [2Fe-2S] cluster is not involved in the dioxygen binding.

Para. 2,

First sentence, I am not sure whether 'architectural elements' was optimal.

Some works have been done to illustrate the possible mechanism why the same RHOs were involved in different reactions. The authors may want to clarify what is known and what is unknown about that question.

Para. 3, it makes me a little confused what have been studied and what not. I was expecting that the authors would show us what is done in this study after a sentence starting with 'however' in Para. 2. However, I found another 'however' in this paragraph.

Para 4

What did 'a polar functional group handle' mean? The sulfonate group? I noticed that a nitro group did not work. So a polar functional group handle did not always work. The authors should exactly define the interaction between protein and substrate.

The sentence starting with 'This diversity in reactivity', it was not convincing. It has been known that NDO could catalyze the monooxygenation and dioxygenation reactions of different substrates.

Section 'Formation of a dioxygenated product by TsaM', the product may be a sequential monooxygenation one, as the authors claimed in the following paragraph.

Figure 3. In the following paragraph and Figure 4, two different in vitro systems, TsaM with H₂O₂ or with VanB-O₂ have different outcomes. Therefore, it was suggested to indicate the electron donor and/or acceptor(s), like did in Figure 2 and 4.

Figure 4, b and c, I am curious why 37 and 38 were oxidized to 41 and 32, but not to 39 and 40, respectively. Even in a single turnover reaction system, 37 could be oxidized to 41 by one enzyme molecule and then to 39 by another enzyme molecule. Insights into the detail mechanism may provide us more interesting information.

And why were the relative intensities shown here?

Panel d, did the ratio of relative intensities of 41 and 39 (as well as 40 and 32) reflect the ratio of their amounts? If so, it should be paid attention and a detail kinetics analysis is necessary. If not, the amount ratio of two products is required. Otherwise, the data did not support a sequential reaction or a dioxygenation one. I preferred that panel d indicated a sequential monooxygenation reaction.

The reaction time in panel d is about 3 hours. And no any control assay was shown. Is it possible that the second oxygenation was non-enzymatic?

Figure 5, residue 195 was not shown in panel b.

Why was TTN shown in panel c, but relative intensity in panel d.

Panel d, the same problems as those in Figure 4b: 1) the ratio of relative intensities did not support a sequential reaction, 2) such a long reaction time implied the possibility of non-enzymatic reaction.

Reviewer #2 (Remarks to the Author):

Review of manuscript NCOMMS-22-46516

Content summary

Tian et al. studied how active site structure modifications determine the type of oxygenation reaction catalyzed by Rieske non-heme ferrous iron oxygenases (ROs), an important class of enzymes in both catabolic and biosynthetic processes. The authors thoroughly evaluated kinetics and product formation of the homotrimeric RO 4-toluenesulfonate methyl-monooxygenase (TsaM) and a suite of mutant enzymes with a comprehensive set of structurally related sulfonated and carboxylated aromatic substrates.

The results of this manuscript show (1) that these enzymes can catalyze different types of oxygenation of aromatic and aliphatic C atoms, namely through dioxygenation, monooxygenation, and sequential monooxygenations. (2) The type of oxygenation reaction correlates with the distance between the oxygenated C atom of the substrate and the non-heme Fe atom in the RO's active site (where dioxygenations require the closest Fe-C distance whereas sequential monooxygenations are favoured by large Fe-C distances). (3) Finally, the authors exemplify the identified role of active site residues by showing that mutants of vanillate O-demethylase (VanA, a monooxygenase) and phthalate dioxygenase (PDO) can be tuned to carry out both type of oxygenations.

General comments

This study features valuable structural insights for the assessment of oxygenation reactions of ROs. Observations showing that ROs can catalyze both mono- and dioxygenations for numerous aromatic substrates are known for decades (see, e.g., ref 7, <https://doi.org/10.1016/j.jmb.2005.03.052>). It has since been speculated that the preferred path of reaction is modulated by how active site residues determine substrate fit and alignment towards the catalytically active non-heme Fe center. Because

substrates do not interact with or bind to the non-heme Fe of ROs (as opposed to substrates of other non-heme Fe oxygenases), the role of the substrate in the active site pocket of ROs for triggering O₂ activation and guiding the ensuing oxygenation path is still elusive. Tian and coworkers now pinpoint an important aspect of the structure-function relationship in ROs with a comprehensive and systematic evaluation that covers many substrates and enzyme variants.

The authors explicitly motivate their study of active site structures of RO with the goal of tuning the oxygenase's reactivity for mono- and dioxygenation reactions. In addition to identifying the various oxygenation products, Tian et al. refer to (sometimes amazingly low) total turnover numbers (TTN) and, in part, kinetic parameters as measures of reactivity. While formally correct, I feel that the assessment of enzymatic reactivity is somewhat incomplete and would merit consideration (or discussion) of additional, potentially important phenomena pertinent to O₂ activation, the efficiency of substrate turnover, and the catalytic cycle of ROs.

The first aspect is that substrate binding in the active site pocket alone might not be sufficient to lead to its oxygenation. One should consider how substrates determine if, when, and how molecular oxygen is activated by ROs (<https://doi.org/10.1021/acs.biochem.5b00573>). Some aromatic compounds could be suitable substrates based on structural considerations. Yet it remains unclear if the presence and alignment of these substrates towards the non-heme Fe is solely responsible for O₂ activation or whether the type of Fe-oxygen species formed (<https://doi.org/10.1021/jacs.8b01822>) and substrate properties matter too (e.g. in that they facilitate Rieske cluster oxidation).

The second factor is that with some substrates, ROs are likely to activate O₂ in an unproductive fashion and thus partially reduce O₂ to reactive oxygen species (ROS). This somewhat ignored but well-known process of O₂-uncoupling leads to low TTNs despite considerable enzyme activity and, potentially, also to enzyme self-inactivation. I wonder whether such considerations would matter for the purpose of tuning ROs for a desired mechanistic outcome.

We have recently studied this O₂ uncoupling for two heterohexameric ROs which also specifically bind their aromatic substrates through H-bonding to a NO₂ group (NBDO, 2NTDO, <https://doi.org/10.1021/acscatal.2c00383> and <https://doi.org/10.1021/acsenviron.2c00023>) and found that the O₂ uncoupling is even the dominant enzymatic activity when expanding the substrate spectrum. We found that substrate structure and thus potentially substrate fit in the active site rather than electronic substrate properties determine whether substrates are oxygenated vs. whether activated O₂ is merely turned into ROS. Even though I can only speculate about the relevance of O₂ uncoupling for homotrimeric ROs, I would assume that unproductive O₂ activation could have happened in experiments with TsaM variants and therefore modulated the obtained total turnover numbers (as shown e.g. in Figure 3c).

The third factor concerns the catalytic mechanism of ROs and the interpretation of steady-state kinetic data. The 2nd subsection on key substrate recognition points (p4-6) is largely based on the interpretation of turnover numbers, $k(\text{cat})$, and half-saturation constants, $K(\text{m})$, and catalytic efficiencies, $k(\text{cat})/K(\text{m})$ from Table 1 for different substrates. I wonder whether these kinetic parameters can be considered to reflect the oxygenation step of enzyme catalysis. As shown for many ROs, the activation of O_2 and formation of reactive Fe-oxygen species is typically the rate-limiting step of catalysis (<https://doi.org/10.1021/jacs.8b01822>, <https://doi.org/10.1007/s00775-004-0537-0>, <https://doi.org/10.1021/jp4076299>). The data in Table 1, while nevertheless valuable in delineating the reactivity of different enzyme-substrate combinations, might rather reflect the rates of O_2 activation in the presence of a specific substrate than the hydroxylation of its aromatic or aliphatic C atoms.

I very much enjoyed reading this manuscript. In addition to my general suggestions for revisions, I have added some specific comments related to the above points as well as a series of editorial recommendations in the following.

Sincerely,

Thomas B. Hofstetter

Specific comments

(s1) The authors nicely state the relevance of ROs by mentioning 70'000 members of this enzyme family. Yet it is unclear why they chose Tsam as model enzyme system to illustrate the effect of active site structure and substrate fit on oxygenation paths. The authors should justify their enzyme selection more specifically. Apparently, the ability of Tsam to bind sulfonated and carboxylated aromatic compounds seems to be critical for Tsam reactivity but the authors do not specify the interactions and residues for these interactions. Referring to a 'polar functional group handle' is not adequate. The ambiguity regarding the residues for substrate binding is surprising given that the remainder of the manuscript is quite specific on critical residues in the active site.

(s2) The discussion of kinetic parameters on page 4 (2nd par) looks somewhat phenomenological. How can the authors conclude about the substrate positioning from discussing steady-state kinetic parameters from Table 1? As mentioned above, substrate hydroxylation is typically not rate-determining in the catalytic cycle of ROs. Do the authors have contrasting information to show?

(s3) Some data in figures 3b/c would merit additional discussion. The contribution of heteroatom dealkylation (Fig. 3b) is substantially smaller and these reactions exhibit very low TTNs. Do the authors have specific explanations for this behaviour? It would also be helpful if the authors specified the

uncertainty of TTNs. In Figure 3c, there is a difference between 4-methyl-benzoate and other para-alkylated benzoates. However, there is no clear trend of decreasing TTN with increasing substituent size. The authors should discuss these trends.

(s4) The use of Fe-C distances for rationalizing the observed reactivity is an interesting perspective but implies that ROs generally show the same 'susceptibility' towards the distance of a C to be hydroxylated. I am not sure whether this is always the case for ROs and suggest that the authors checked their conclusions on pages 10/11 more critically. Some examples: (1) In NBDO, data suggest that sequential monooxygenations are not likely to happen. First, this enzyme can carry out both mono- and dioxygenations in an apparently simultaneous fashion with the same substrate. An example are the well documented reactions with 2- and 4-nitrotoluenes (ref 7 and <https://doi.org/10.1021/acs.est.5b05084>). Second, monooxygenated aromatic compounds (i.e., substituted phenols (like 4-hydroxybenzoate (32) and 4-hydroxybenzene-sulfonate) cause 100% unproductive O₂ activation in NBDO. It would thus not be possible to find sequential monooxygenations. (2) In naphthalene dioxygenase (NDO), we have shown that the proximity of the substrate could indeed be an important parameter for determining reactivity but rather by facilitating the oxidation of the Rieske cluster (<https://doi.org/10.1002/chem.202103937>). Thus, while the proximity is certainly a key factor, it seems somewhat simplified to interpret this factor as guiding the sequential hydroxylations of ROs in general. See also comments above.

(s5) p9, 2nd par. The observations of aldehyde-type products are really interesting but also puzzling. Oxidation of alcohols to aldehydes (i.e., reactions 1 -> 3 -> 57 and 2 -> 4 -> 58) seems unusual for ROs as one would expect this be a hydride transfer. But I might be wrong. Is there any precedent for this reaction catalyzed by ROs? If so, could the authors elaborate on this (unusual/unexpected) type of catalysis and explain why this is possible?

(s6) p9, 4th par. I concur with the validity and interpretation of these observation. Nevertheless, the TTNs in Suppl. Fig 59 and 60 are quite small. Could this suggest that the enzyme is structurally poised to carry out this reaction but that it is very inefficient due to O₂ uncoupling? The authors could have modulated the amount of added reduction equivalents (NADH) to and establish electron balances. Without reference to low TTNs and a potentially inefficient substrate oxygenation, the interpretation of the observed reactivity here might appear too optimistic.

Editorial comments

(e1) Terminology: (1) Referring to the various substrates almost exclusively by numbers instead of compound names makes the reading of the manuscript quite tedious and overly complicated. I strongly recommend to use compound names wherever possible because these allow for making intuitive chemical comparisons while reading (e.g., by referring to hydroxylated aromatic compounds vs. benzylic alcohols etc.). (2) The attribute 'native' for description of a substrate to which enzymatic function has

been optimized to is not necessarily correct. There are cases in which the substrate name primarily relates to the procedure and conditions with which an enzyme has been identified and characterized. In fact, there are several examples showing that the so-called native substrate can be 'outperformed' in terms of rates and turnover by other compounds. Consider rewording.

(e2) Some questions on chemical terminology. First, the active site Fe is sometimes referred to as mononuclear. This is correct, yet presumably unnecessary because ROs do not exhibit dinuclear Fe in their active sites. Second, is it correct to refer to an aromatic methyl substituent as 'exocyclic'? I don't think so.

(e3) The comparison of relative intensities (e.g., y-axes in Figures 2b, 4c/d,) without specification of their meaning (relative to what) could be misleading. In Figure 4d, for example, it would be much more insight full to learn if the concentrations of 39 and 41 match the initial amount of 37. Instead, total relative intensities at times exceed 100%.

(e4) p3, 3rd paragraph (par). Should read '18O-labelled and unlabelled H₂O₂'.

(e5) p4, 2nd par. The logic of discussion of kinetic parameters is unclear. $k(\text{cat})$ of substances 1 and 2 differ by only 12%, the ratio of $k(\text{cat})/K(\text{m})$ is only by a factor of 3.2. What is the meaning of this comparison?

(e6) p4, 2nd par. What is an 'architectural feature' in this context? This subsection rather looks like a study of substrate specificity.

(e7) p4 and throughout manuscript. Consider substituting 'charged handle' with a biochemically precise description of the substrate binding site.

(e8) p5, 1st and 2nd par. The numbers on catalytic efficiencies and K_m in text do not seem fully consistent with those of Table 1 (930±95 for 4-ethyl-benzoate). Please revise.

(e9) p6, 1st par. What do the authors refer to as 'similar size'?

(e10) p6, 1st par. "...these results highlight the importance of the identity of the functional group at the p-position of the substrate to the efficiency of the TsaM-catalyzed monooxygenation". This is a very well

known phenomenon and the authors could provide more references (and credit) to earlier works (e.g., ref 7 and others).

(e11) p6, 2nd par vs. 3rd par. First, the consequences of removing the 4-alkyl substituent and an apparent decrease of reactivity is discussed. Shortly thereafter (next par), the authors reveal that the reaction was carried on to a catecholic product. This is rather misleading. I suggest that the authors convey the full interpretation right away without 'mislead' readers first.

(e12) p7, 1st par. The ability of TsAM and CAO to perform sequential monooxygenations is interesting but it might not apply to other ROs. Why would this comparison be particularly insightful?

(e13) p7, 1st par. Why should the acidity of C-H bonds matter? Wouldn't they be in the range of >15 and higher and far from 'acidic'?

(e14) p7, 2nd par. This is difficult to read. If the whole purpose of this subsection was to start thinking about making sequential monooxygenases into dioxygenase, why not telling right away?

(e15) p7, 3rd par. 'Substrate electronics' is not a proper description of chemical properties and I feel that this hypothesis does not make sense. Even strong electron donating moieties of substrates do not necessarily make reactions with electrophilic Fe-oxygen go better or faster given that O₂ activation could be the overall rate-limiting step of catalysis.

(e16) Figure 5a: Why did the authors not provide references to the corresponding papers? PDB ID are fine but access to the original numbers (in the cited papers) would be even more insightful to readers.

(e17) p9, 3rd and 4th par. It would be interesting to have some visualization of the mutant enzymes active site structures for VanA and PDO. Could a reference to a modified Suppl. Fig. 38 help for this purpose?

Reviewer #3 (Remarks to the Author):

The manuscript by Tian et al. outlines the investigation of architectural features that dictate monooxygenation, sequential monooxygenation or dioxygenation reactions in the two-component Rieske oxygenase TsaM, and two additional representatives (VanA and PDO). The analysis was initiated by a thorough 'substrate engineering' approach comprising a wide range of substrates rationally chosen to characterize the catalytic ability of TsaM to catalyze monooxygenation, sequential monooxygenation, and even desaturation and dealkylation reactions. By performing rational protein design targeting several residues in the active site of TsaM, it was possible to identify several important amino acid residues that govern the catalytic outcome in this enzyme. Subsequently, the identified architectural features were transferred to VanA and PDO, two related Rieske's characterized as monooxygenase and dioxygenase, respectively. Evaluation of reaction outcome, created active site models and mutational studies were used to explain and validate findings and establish broader impact of the study.

The manuscript is scientifically sound and well-presented with a solid supporting information and methods part.

Major strengths and weaknesses of the present paper:

- The systematic analysis (and presentation in the manuscript) of the structural features that govern the reaction outcome in the protein family of Rieske oxygenases are positively noteworthy. The obtained results and interpretation thereof clearly contribute to increased knowledge and understanding of how these complex enzymes achieve different reaction outcomes (based on active site and substrate architecture).
- On the other hand, a demonstrated ability to predict or custom-tune monooxygenation / sequential monooxygenation over dioxygenation might not be fully supported by the study. The reason is that a full switch to monooxygenation in PDO by introducing the mutation I232G did not furnish the enzyme into an exclusive monooxygenase. Similarly, TsaM could not be fully engineered switched into a dioxygenase. As such, the herein-described principles of substrate architecture and proximity to the iron are accompanied by other factors that govern reaction outcomes that are of less 'predictive' nature.
- In addition, residues found within the active site of the alpha-subunit have been shown before to play key roles in dictating the site-, stereo-, and chemoselectivity of a Rieske oxygenase-catalyzed reaction. For instance, it has been shown that in carbazole dioxygenase several mutations in the active site can change the orientation in which the substrate binds, and thus the outcome of the hydroxylation event (compare: Ref 20: Inoue K, et al. *Appl Environ Microbiol* 2014, 80:2821–2832).

Other studies of naphthalene dioxygenase also revealed that changes in the active site affect the reaction outcome. For instance, some form a monooxygenated product, rather than dioxygenated products using non-native substrates (compare: Ref. 54: Parales et al., *J Bacteriol* 2000, 182:1641–1649; Ref 57: Yu et al., *J Ind Microbiol Biotechnol* 2001, 27:94–103; Ref 32: Halder et al, *ChemCatChem* 2018, 10:178–182).

As such, the approach for studying the active site architecture and the reaction scope when using different non-natural substrates is based on these earlier studies. Conceptually, the ability to transfer

mutations that influence reaction outcomes to closely related Rieske's is a novel aspect that strengthens the work.

Other suggestions:

- The reported kinetic parameters (Table 1) for the different substrates require further clarification. Typically, k_{cat} values should be based on the number of active sites ($n=3?$) per trimer of the Oxy. The authors should clarify whether if this has been taken into account and indicate this in the methods part.
- In the a3 architecture such as in TsaM, the subunit-subunit interfaces are the site where the partner reductase bind. For TsaM as well as for PDO, the 'non-native' redox partner VanB has been used. Since the binding of the redox partner induces several conformational changes to facilitate electron transfer, a non-native redox partner might also influence the reaction outcome. As for TsaM the native reductase TsaB could not be functionally reconstituted in vitro, I suggest performing a control experiment with PDO and its native reductase PDR for the wild-type PDO and the variant I256G to exclude an effect of the non-native redox partner on the reaction outcome.
- Another point is the dependency of oxygen on the reaction outcome (mono- vs. dihydroxylation). Did the authors observe any effect of oxygen on the reaction outcome, e.g. by determining the apparent K_m for oxygen?

Reviewer #1:

Tian et al. describes the Rieske oxygenase reactivity of TsaM from *Comamonas testosteroni*. They found TsaM could catalyze a variety of different reactions, including monooxygenation, and sequential monooxygenation. They considered the distance between substrate molecules and iron atoms are the key for the reactions and tune the reaction activity of TsaM/VanA via site-directed mutagenesis. Some interesting findings were observed, including different oxygenation reactions in different enzymatic systems and the tuning of VanA reactivity via mutagenesis. However, I have some concerns.

1. Data in figure 4d and 5d is not convincing. a) Only relative intensity was shown, not supporting the products generated from a sequential oxygenation, and b) the reaction time was too long to exclude non-enzymatic oxygenation.

We thank Reviewer 1 for the comment. We replotted the data in panels 4d and 5d as product concentration over time instead of using relative intensity. We also added a control reaction that is plotted in the same manner over the same time scale. In this control reaction, the enzyme was excluded but all other reaction components were present. We did not observe any oxygenation in this control. This control experiment can be found as Supplementary Figures 36-37 and 63. The TTNs for the data in Figure 4d are included in Supplementary Figure 38.

2. A model is suggested to show why the reactivity tuning could be achieved. For now, only the distance between substrate molecules and iron atoms were mentioned.

We thank Reviewer 1 for this comment. We added additional description to the discussion to explain our model and now account for not only the different types of chemistry observed, but also the different levels of activity that were measured.

3. The manuscript could be improved after a detail check, especially for Introduction.

We thank Reviewer 1 for the comment. We carefully checked over the manuscript for errors.

Abstract

Please clearly show your contribution. More technologic details should be helpful.

We updated the abstract to be more concise about our contribution.

Introduction

Para. 1,

Sentence starting with "For example", references should be cited.

We thank Reviewer 1 for pointing out this omission. We have now added appropriate references for this statement.

Sentence starting with 'Collectively', the Rieske [2Fe-2S] cluster is not involved in the dioxygen binding.

Para. 2,

We rewrote this section and have changed this sentence completely.

First sentence, I am not sure whether 'architectural elements' was optimal.

We updated the sentence to read: "Likewise, the structural motifs that Rieske oxygenases employ to facilitate their different catalytic outcomes remain unclear."

Some works have been done to illustrate the possible mechanism why the same RHOs were involved in different reactions. The authors may want to clarify what is known and what is unknown about that question.

We added more details into the introduction regarding prior work on CARDO, NDO, CDO, and NBDO.

Para. 3, it makes me a little confused what have been studied and what not. I was expecting that the authors would show us what is done in this study after a sentence starting with 'however' in Para. 2. However, I found another 'however' in this paragraph.

We removed the two instances of however in these paragraphs for clarification.

Para 4:

What did 'a polar functional group handle' mean? The sulfonate group? I noticed that a nitro group did not work. So a polar functional group handle did not always work. The authors should exactly define the interaction between protein and substrate.

We updated this text to read: "Through this analysis, we demonstrate that TsaM shows a preference for performing monooxygenation reactions on substrates that contain a polar functional group on the C1 carbon and functionalizes the methyl group or the benzylic carbon of substrates with longer carbon chains."

As described below, p-nitrotoluene is a substrate of TsaM and does get functionalized by the enzyme to a small extent (please see the heat map in Figure 3a, bar graph in Supplementary Fig. 17, and apparent kinetic parameters in Table 1 and Supplementary Fig. 10). Through performing the uncoupling assays suggested by Reviewer 2, it was determined that the low turnover is due to an increased amount of uncoupling when p-nitrotoluene is used as a substrate relative to p-toluenesulfonate and 4-methylbenzoate (see Fig. 3f and Supplementary Fig. 15). Additionally, we added a model of how the substrate interacts with TsaM to Figure 3c.

The sentence starting with 'This diversity in reactivity', it was not convincing. It has been known that NDO could catalyze the monooxygenation and dioxygenation reactions of different substrates.

We rewrote this section and have changed this sentence completely.

Section 'Formation of a dioxygenated product by TsaM', the product may be a sequential monooxygenation one, as the authors claimed in the following paragraph.

*We thank Reviewer 1 for pointing out that this section title was misleading in the context that we are showing both sequential oxygenation and dioxygenation chemistry in this manuscript. Therefore, we changed this section title to be: **Formation of a sequentially oxygenated product by TsaM.***

Figure 3. In the following paragraph and Figure 4, two different in vitro systems, TsaM with H₂O₂

or with VanB-O2 have different outcomes. Therefore, it was suggested to indicate the electron donor and/or acceptor(s), like did in Figure 2 and 4.

We thank Reviewer 1 for this comment. We have clarified the Figures by adding the identity of the electron donor in Figure 3d, Figure 3e, Figure 5c-d, and Figure 6a-b.

Figure 4, b and c, I am curious why 37 and 38 were oxidized to 41 and 32, but not to 39 and 40, respectively. Even in a single turnover reaction system, 37 could be oxidized to 41 by one enzyme molecule and then to 39 by another enzyme molecule. Insights into the detail mechanism may provide us more interesting information.

We hypothesize that we do not see the second oxidation to 39 and 40 because of the way that our reactions were run. First, as mentioned by the reviewer, we are operating under single turnover conditions meaning that 37 and 38 were oxidized to 41 and 32, respectively. Second, the reason we do not see conversion of 37 and 38 into 39 and 40 is that our starting concentrations of 37 and 38 are much higher than what would be produced of 41 and 32 in a single turnover (e.g. we add 2 mM substrate to the reaction that contains 10 uM enzyme). Further, it has been shown for several Rieske oxygenases that product is not released without re-reduction of the mononuclear iron center and that is why the peroxide shunt reaction leads to only a single turnover (J. Biol. Chem. 2003, 278, 2, 829-35).

And why were the relative intensities shown here?

We thank Reviewer 1 for the comment. As described above, we replotted the data in panel 4d as product concentration over time instead of using relative intensity. We also added a control reaction that is plotted in the same manner over the same time scale. In this control reaction, the enzyme was excluded but all other reaction components were present. We did not observe any oxygenation in this control. These control experiments can also be found in Supplementary Figures 36-37. The TTN for the data in Figure 4d are included in Supplementary Figure 38.

Panel d, did the ratio of relative intensities of 41 and 39 (as well as 40 and 32) reflect the ratio of their amounts? If so, it should be paid attention and a detail kinetics analysis is necessary. If not, the amount ratio of two products is required. Otherwise, the data did not support a sequential reaction or a dioxygenation one. I preferred that panel d indicated a sequential monooxygenation reaction.

We thank Reviewer 1 for the comment. As described above, we now plotted this data as product concentration over time. We also added two figures into the SI that show the ratios of the products that contain one versus two incorporated oxygen atoms over time. From this figure, you will see that the total product concentration saturates. Associated data is now incorporated into Supplementary Figures 36-37.

The reaction time in panel d is about 3 hours. And no any control assay was shown. Is it possible that the second oxygenation was non-enzymatic?

We thank Reviewer 1 for the comment. As described above, we have added a control reaction that is also plotted as product concentration over time. In this control reaction, the enzyme was excluded, but all other reaction components were present. We did not observe any oxygenation in this control. This control experiment can also be found plotted in Supplementary Figures 36-37.

Figure 5, residue 195 was not shown in panel b.

We thank Reviewer 1 for pointing out the absence of this residue. We now realize that we had an error in the residue numbering but have updated the residues and their numbers in the text as well as in Figure 5 and Supplementary Figure 50.

Why was TTN shown in panel c, but relative intensity in panel d. Panel d, the same problems as those in Figure 4b: 1) the ratio of relative intensities did not support a sequential reaction, 2) such a long reaction time implied the possibility of non-enzymatic reaction.

We thank Reviewer 1 for the comment. We have now replotted the data in panel 5d as product concentration over time instead of using relative intensity. We also added a control reaction to Figure 5d that is plotted in the same manner over the same time scale. In this control reaction, the enzyme was excluded but all other reaction components were present. We did not observe any oxygenation in this control. This control experiment can be found with the new data that is plotted in Supplementary Figure 63.

Reviewer #2:

Content summary:

Tian et al. studied how active site structure modifications determine the type of oxygenation reaction catalyzed by Rieske non-heme ferrous iron oxygenases (ROs), an important class of enzymes in both catabolic and biosynthetic processes. The authors thoroughly evaluated kinetics and product formation of the homotrimeric RO 4-toluenesulfonate methyl-monoxygenase (TsaM) and a suite of mutant enzymes with a comprehensive set of structurally related sulfonated and carboxylated aromatic substrates.

The results of this manuscript show (1) that these enzymes can catalyze different types of oxygenation of aromatic and aliphatic C atoms, namely through dioxygenation, monooxygenation, and sequential monooxygenations. (2) The type of oxygenation reaction correlates with the distance between the oxygenated C atom of the substrate and the non-heme Fe atom in the RO's active site (where dioxygenations require the closest Fe-C distance whereas sequential monooxygenations are favoured by large Fe-C distances). (3) Finally, the authors exemplify the identified role of active site residues by showing that mutants of vanillate O-demethylase (VanA, a monooxygenase) and phthalate dioxygenase (PDO) can be tuned to carry out both type of oxygenations.

General comment:

This study features valuable structural insights for the assessment of oxygenation reactions of ROs. Observations showing that ROs can catalyze both mono- and dioxygenations for numerous aromatic substrates are known for decades (see, e.g., ref 7, <https://doi.org/10.1016/j.jmb.2005.03.052>). It has since been speculated that the preferred path of reaction is modulated by how active site residues determine substrate fit and alignment towards the catalytically active non-heme Fe center. Because substrates do not interact with or bind to the non-heme Fe of ROs (as opposed to substrates of other non-heme Fe oxygenases), the role of the substrate in the active site pocket of ROs for triggering O₂ activation and guiding the ensuing oxygenation path is still elusive. Tian and coworkers now pinpoint an important aspect of the

structure-function relationship in ROs with a comprehensive and systematic evaluation that covers many substrates and enzyme variants.

The authors explicitly motivate their study of active site structures of RO with the goal of tuning the oxygenase's reactivity for mono- and dioxygenation reactions. In addition to identifying the various oxygenation products, Tian et al. refer to (sometimes amazingly low) total turnover numbers (TTN) and, in part, kinetic parameters as measures of reactivity. While formally correct, I feel that the assessment of enzymatic reactivity is somewhat incomplete and would merit consideration (or discussion) of additional, potentially important phenomena pertinent to O₂ activation, the efficiency of substrate turnover, and the catalytic cycle of ROs.

We thank Reviewer 2 for making this point and we have added additional discussion into the introduction regarding what is known about Rieske oxygenase chemistry. Additionally, we amplified the activity of the TsaM catalyzed dioxygenation reaction by creation of new rational variants, and by inclusion of the non-native reductases that work with PDO. Further, we added text regarding how to interpret the low turnover numbers and kinetic parameters of TsaM into the discussion.

The first aspect is that substrate binding in the active site pocket alone might not be sufficient to lead to its oxygenation. One should consider how substrates determine if, when, and how molecular oxygen is activated by ROs (<https://doi.org/10.1021/acs.biochem.5b00573>). Some aromatic compounds could be suitable substrates based on structural considerations. Yet it remains unclear if the presence and alignment of these substrates towards the non-heme Fe is solely responsible for O₂ activation or whether the type of Fe-oxygen species formed (<https://doi.org/10.1021/jacs.8b01822>) and substrate properties matter too (e.g. in that they facilitate Rieske cluster oxidation).

The second factor is that with some substrates, ROs are likely to activate O₂ in an unproductive fashion and thus partially reduce O₂ to reactive oxygen species (ROS). This somewhat ignored but well-known process of O₂-uncoupling leads to low TTNs despite considerable enzyme activity and, potentially, also to enzyme self-inactivation. I wonder whether such considerations would matter for the purpose of tuning ROs for a desired mechanistic outcome.

We thank Reviewer 2 for making this point and we have added additional experiments to probe O₂ uncoupling for every substrate, variant, and reductase combination investigated in this work (see discussion below). We also have the above cited references included in the text.

We have recently studied this O₂ uncoupling for two heterohexameric ROs which also specifically bind their aromatic substrates through H-bonding to a NO₂ group (NBDO, 2NTDO, <https://doi.org/10.1021/acscatal.2c00383> and <https://doi.org/10.1021/acsenvronau.2c00023>) and found that the O₂ uncoupling is even the dominant enzymatic activity when expanding the substrate spectrum. We found that substrate structure and thus potentially substrate fit in the active site rather than electronic substrate properties determine whether substrates are oxygenated vs. whether activated O₂ is merely turned into ROS. Even though I can only speculate about the relevance of O₂ uncoupling for homotrimeric ROs, I would assume that unproductive O₂ activation could have happened in experiments with TsaM variants and therefore modulated the obtained total turnover numbers (as shown e.g. in Figure 3c).

As described above, we performed an oxygen uncoupling assay for:

1. Each of the 17 TsaM-catalyzed reactions performed in Figure 3 in triplicate. This data is plotted in Figure 3f. Additional details concerning the experimental setup for this reaction are included in Supplementary Figure 15.

Additionally, to improve the rigor of this work, we performed the oxygen uncoupling assay in triplicate for each of the following:

2. TsaM with benzenesulfonate, benzoate, aniline, phenol and this data is incorporated into Supplementary Figure 43.
3. TsaM with N-phenylacetamide, phenylacetate, or 4-methylphenylacetate, benzoylformate, and 4-methylbenzoylformate and this data is incorporated into Supplementary Figure 43.
4. Each of the TsaM variants (M230F, T232F, M230W, T232W, T232I, M230W/T232I, M230W/T232I/S257R, and M230W/T232I/S257R/Y269V) with a benzoate substrate and this data is incorporated into Supplementary Figure 62.
5. The M230G/T232G TsaM variant that leads to sequential monooxygenation and this data is incorporated into Supplementary Figure 62.
6. All of the wild-type and variant PDO assays, and this data is incorporated into Supplementary Figure 72.

From this list of experiments, only *p*-nitrotoluene, *p*-isopropyltoluene, phenylacetate, and 4-methylphenylacetate showed production of H₂O₂ above that produced by TsaM with *p*-toluenesulfonate or 4-methylbenzoate substrates. Therefore, we also performed intact mass spectrometry experiments on these reactions and did not see any modification. This data is incorporated into Supplementary Figure 16. For Supplementary Figure 16, intact mass spectrometry experiments for the TsaM variants that give rise to the highest (M230W/T232I/S257R/Y269V) and lowest (T232I) amounts of dioxygenated product. These experiments did not reveal any protein modification. The references suggested are also included in the text.

The third factor concerns the catalytic mechanism of ROs and the interpretation of steady-state kinetic data. The 2nd subsection on key substrate recognition points (p4-6) is largely based on the interpretation of turnover numbers, $k(\text{cat})$, and half-saturation constants, $K(\text{m})$, and catalytic efficiencies, $k(\text{cat})/K(\text{m})$ from Table 1 for different substrates. I wonder whether these kinetic parameters can be considered to reflect the oxygenation step of enzyme catalysis. As shown for many ROs, the activation of O₂ and formation of reactive Fe-oxygen species is typically the rate-limiting step of catalysis (<https://doi.org/10.1021/jacs.8b01822>, <https://doi.org/10.1007/s00775-004-0537-0>, <https://doi.org/10.1021/jp4076299>). The data in Table 1, while nevertheless valuable in delineating the reactivity of different enzyme-substrate combinations, might rather reflect the rates of O₂ activation in the presence of a specific substrate than the hydroxylation of its aromatic or aliphatic C atoms.

We thank Reviewer 2 for this comment. As described below, we have added more background information regarding what is known about Rieske oxygenase catalysis into the introduction. We have further indicated that we are reporting apparent kinetic parameters and have clarified the language in the text. We also have all three papers referenced in the manuscript.

I very much enjoyed reading this manuscript. In addition to my general suggestions for revisions,

I have added some specific comments related to the above points as well as a series of editorial recommendations in the following.

We thank Reviewer 2 for this positive assessment of our work, and we anticipate that that you will find the revised manuscript to be even more rigorous in the science and overall, more impactful to the community.

Sincerely,
Thomas B. Hofstetter

Specific comments

(s1) The authors nicely state the relevance of ROs by mentioning 70'000 members of this enzyme family. Yet it is unclear why they chose TsaM as model enzyme system to illustrate the effect of active site structure and substrate fit on oxygenation paths. The authors should justify their enzyme selection more specifically. Apparently, the ability of TsaM to bind sulfonated and carboxylated aromatic compounds seems to be critical for TsaM reactivity but the authors do not specify the interactions and residues for these interactions. Referring to a 'polar functional group handle' is not adequate. The ambiguity regarding the residues for substrate binding is surprising given that the remainder of the manuscript is quite specific on critical residues in the active site.

We thank Reviewer 2 for this comment. First, we added additional details regarding why TsaM was chosen for this work into the introduction.

Second, we added a model of how the substrate interacts with TsaM to Figure 3c and Supplementary Figure 12. We made this model by overlaying the substrate bound structure of DdmC with an AlphaFold model of TsaM. Since dicamba has a carboxylate functional group, we were able to identify potential interacting residues in the active site of TsaM. DdmC shares 35-percent sequence identity with TsaM and the active sites are comparable in that there are only a few clustered polar residues that would be sufficient to recognize a carboxylate or sulfonate moiety.

We also altered the introductory sentence that referred to substrate recognition in the introduction to read: "Through this analysis we demonstrate that TsaM shows a preference for performing monooxygenation reactions on substrates that contain a polar functional group and functionalizes the methyl group or the benzylic carbon of substrates with longer carbon chains."

(s2) The discussion of kinetic parameters on page 4 (2nd par) looks somewhat phenomenological. How can the authors conclude about the substrate positioning from discussing steady-state kinetic parameters from Table 1? As mentioned above, substrate hydroxylation is typically not rate-determining in the catalytic cycle of ROs. Do the authors have contrasting information to show?

We agree with Reviewer 2 that we could have included more details regarding the kinetic parameters. As such, we have included more discussion regarding what is known about Rieske oxygenase catalyzed reactions into the introduction. We also added a statement that reads "As previously described, in this work, for all substrates tested, we report the apparent kinetic parameters because the saturating concentrations of NADH and O₂ in this TsaM-VanB system have not been determined". We also clarified the language that inappropriately conveyed that we learned about substrate positioning from the measured values. Finally, we address the proposed causes of low measured activity in the discussion.

(s3) Some data in figures 3b/c would merit additional discussion. The contribution of heteroatom dealkylation (Fig. 3b) is substantially smaller and these reactions exhibit very low TTNs. Do the authors have specific explanations for this behaviour? It would also be helpful if the authors specified the uncertainty of TTNs. In Figure 3c, there is a difference between 4-methylbenzoate and other para-alkylated benzoates. However, there is no clear trend of decreasing TTN with increasing substituent size. The authors should discuss these trends.

We thank Reviewer 2 for these comments. To address your first point, we added a description regarding the different activities observed with p-(methoxy)benzoate, p-(methylamino)benzoate, and p-(methylthio)benzoate into the discussion. To complement our hypothesis that the turnover numbers of TsaM with these molecules is low due to a decreased ability of the enzyme to position these molecules in a productive catalytic orientation, we added a Supplementary Figure to detail the different charge distributions relative to p-toluenesulfonate and 4-methylbenzoate (Supplementary Figure 34). We also want to mention a manuscript that we recently submitted. This manuscript describes how to engineer TsaM to accept such a substrate, behave as DdmC or VanA, and perform an oxidative dealkylation reaction on dicamba, vanillate, o-methylbenzoate, or m-methylbenzoate. In this work, we determined that mutations needed to be introduced into our so-called “hotspot” regions, the active site, substrate entrance tunnel, and flexible connecting loop, to confer this functionality to TsaM. When these six-to-ten needed mutations are introduced into TsaM, the TTNs of TsaM on these molecules is no longer significantly different from that observed with DdmC or VanA.

Second, to show the uncertainty in the TTNs that are illustrated in Figure 3a-3b as a heat map, we added a Supplementary Figure with error bars (Supplementary Figure 17).

Third, to address your comment regarding 4-methylbenzoate and the other para-alkylated benzoates, we rearranged the way the data was represented in Figure 3d. In this rearrangement, the trend of decreasing activity with increasing functional group length is clear. We also added the following statement to the text to describe the observed trend: “despite this additional reactivity, from these experiments a clear trend can be defined: increasing the substituent length at the para-position from a methyl to an ethyl, propyl, butyl, or pentyl functional group is correlated with lower total turnover numbers (Figure 3d). The 4-isopropylbenzoate substrate is an outlier from this trend, presumably due to the branched, rather than linear nature, of the isopropyl functional group (Figure 3d).”

(s4) The use of Fe-C distances for rationalizing the observed reactivity is an interesting perspective but implies that ROs generally show the same ‘susceptibility’ towards the distance of a C to be hydroxylated. I am not sure whether this is always the case for ROs and suggest that the authors checked their conclusions on pages 10/11 more critically. Some examples: (1) In NBDO, data suggest that sequential monooxygenations are not likely to happen. First, this enzyme can carry out both mono- and dioxygenations in an apparently simultaneous fashion with the same substrate. An example are the well documented reactions with 2- and 4-nitrotoluenes (ref 7 and <https://doi.org/10.1021/acs.est.5b05084>). Second, monooxygenated aromatic compounds (i.e., substituted phenols (like 4-hydroxybenzoate (32) and 4-hydroxybenzenesulfonate) cause 100% unproductive O₂ activation in NBDO. It would thus not be possible to find sequential monooxygenations. (2) In naphthalene dioxygenase (NDO), we have shown that the proximity of the substrate could indeed be an important parameter for determining reactivity but rather by facilitating the oxidation of the Rieske cluster (<https://doi.org/10.1002/chem.202103937>). Thus, while the proximity is certainly a key factor, it seems somewhat simplified to interpret this

factor as guiding the sequential hydroxylations of ROs in general. See also comments above.

We thank Reviewer 2 for these helpful references which are incorporated into the manuscript.

(s5) p9, 2nd par. The observations of aldehyde-type products are really interesting but also puzzling. Oxidation of alcohols to aldehydes (i.e., reactions 1 -> 3 -> 57 and 2 -> 4 -> 58) seems unusual for ROs as one would expect this be a hydride transfer. But I might be wrong. Is there any precedent for this reaction catalyzed by ROs? If so, could the authors elaborate on this (unusual/unexpected) type of catalysis and explain why this is possible?

We thank Reviewer 2 for this comment, we recently published a manuscript about the chemistry of CAO (ACS Cent Sci. 2022, 8, 10, 1393–1403) and showed that both oxygenation reactions, the conversion of a methyl group into a hydroxymethyl group, and the subsequent formation of the Chlorophyllide b formyl group require Rieske oxygenase chemistry. As described below, another example of this type of chemistry is catalyzed by the Rieske oxygenase PrnD (Angew. Chem. Int. Ed. 2006, 45 (4), 622-625). We posit that two examples and a multitude of uncharacterized Rieske oxygenases might suggest we will see more of this chemistry in the future.

(s6) p9, 4th par. I concur with the validity and interpretation of these observation. Nevertheless, the TTNs in Suppl. Fig 59 and 60 are quite small. Could this suggest that the enzyme is structurally poised to carry out this reaction but that it is very inefficient due to O₂ uncoupling? The authors could have modulated the amount of added reduction equivalents (NADH) to and establish electron balances. Without reference to low TTNs and a potentially inefficient substrate oxygenation, the interpretation of the observed reactivity here might appear too optimistic.

We thank Reviewer 2 for this comment as it inspired us, as described above, to investigate uncoupling in the reactions performed with every substrate, variant protein, and reductase combination used in this work. Although in many cases, we did not see a significant increase in uncoupling relative to that observed with TsaM/VanB or the other wild-type system (PDO/PDR), these reactions did provide insight into why only a low level of activity was observed with a p-nitrotoluene substrate and did reveal that combination of TsaM with p-isopropyltoluene leads to 100-percent unproductive oxygen activation.

The uncoupling assay measurements that specifically pertain to the dioxygenation variant data is shown in Supplementary Figures 43, 62, and 72. To complement this data, intact mass spectrometry data is also included in Supplementary Figure 16.

Editorial comments

(e1) Terminology: (1) Referring to the various substrates almost exclusively by numbers instead of compound names makes the reading of the manuscript quite tedious and overly complicated. I strongly recommend to use compound names wherever possible because these allow for making intuitive chemical comparisons while reading (e.g., by referring to hydroxylated aromatic compounds vs. benzylic alcohols etc.). (2) The attribute 'native' for description of a substrate to which enzymatic function has been optimized to is not necessarily correct. There are cases in which the substrate name primarily relates to the procedure and conditions with which an enzyme has been identified and characterized. In fact, there are several examples showing that the so-called native substrate can be 'outperformed' in terms of rates and turnover by other compounds. Consider rewording.

We thank Reviewer 2 for this helpful comment. Whereas we included the numbers with the first reference to a compound, we substituted the compound names for almost all other occurrences.

We also removed most cases of native and if it remains, we clarified that it is the “reported native” substrate in the literature.

(e2) Some questions on chemical terminology. First, the active site Fe is sometimes referred to as mononuclear. This is correct, yet presumably unnecessary because ROs do not exhibit dinuclear Fe in their active sites. Second, is it correct to refer to an aromatic methyl substituent as ‘exocyclic’? I don’t think so.

We thank Reviewer 2 for this helpful comment. We removed the redundant occurrences of mononuclear in this manuscript. We originally decided to use the term exocyclic because it is similarly used to describe the reaction catalyzed by DdmC and differentiate this enzyme from the canonical Rieske oxygenase enzymes that perform chemistry on an aromatic ring (see D’Ordine and Dumitru et. al.). However, for clarity to the reader, we removed most instances of this term and clarified its meaning if present.

(e3) The comparison of relative intensities (e.g., y-axes in Figures 2b, 4c/d,) without specification of their meaning (relative to what) could be misleading. In Figure 4d, for example, it would be much more insight full to learn if the concentrations of 39 and 41 match the initial amount of 37. Instead, total relative intensities at times exceed 100%.

We thank Reviewer 2 for the comment. For Figure 2, we replotted the data as ion count versus retention since this is the mass spectrometry data. We added Supplementary Figure 6 to contain TTNs for each tested reductase shown in Figure 2b. In addition, as described above, we replotted the data in panels 4d and 5d as product concentration over time instead of using relative intensity. Additionally, we added bar graphs into Supplementary Figures 36, 37, and 63 to show the ratio of the products formed and that the total amount of the products formed saturates over time. Further, we added TTN for the data in Figure 4d into Supplementary Figure 38.

(e4) p3, 3rd paragraph (par). Should read ‘¹⁸O-labelled and unlabelled H₂O₂’.

We updated the text as suggested.

(e5) p4, 2nd par. The logic of discussion of kinetic parameters is unclear. $k(\text{cat})$ of substances 1 and 2 differ by only 12%, the ratio of $k(\text{cat})/K(\text{m})$ is only by a factor of 3.2. What is the meaning of this comparison?

We updated this section of the manuscript to convey our logic and it now reads: “As a foundation for investigating the reactivity of the TsaM-VanB system and delineating reactivity with different substrate molecules, a steady-state kinetic analysis was performed on the reported native substrates⁴⁴⁻⁴⁶”.

(e6) p4, 2nd par. What is an ‘architectural feature’ in this context? This subsection rather looks like a study of substrate specificity.

We updated the text to read: “...a study was undertaken to determine the substrate specificity of TsaM.”

(e7) p4 and throughout manuscript. Consider substituting 'charged handle' with a biochemically precise description of the substrate binding site.

We updated the text to be more specific in each of the instances where handle was used.

(e8) p5, 1st and 2nd par. The numbers on catalytic efficiencies and K_m in text do not seem fully consistent with those of Table 1 (930 ± 95 for 4-ethyl-benzoate). Please revise.

We updated the numbers to be consistent with the significant figures included in Table 1.

(e9) p6, 1st par. What do the authors refer to as 'similar size'?

The sentence has been updated to read: "Despite the structural resemblance..."

(e10) p6, 1st par. "...these results highlight the importance of the identity of the functional group at the p-position of the substrate to the efficiency of the Tsam-catalyzed monooxygenation". This is a very well known phenomenon and the authors could provide more references (and credit) to earlier works (e.g., ref 7 and others).

Since we added a lot of new data to this section, it has been substantially revised. In the discussion where we talk about the importance of this functional group (end of the third paragraph) we added references as suggested.

(e11) p6, 2nd par vs. 3rd par. First, the consequences of removing the 4-alkyl substituent and an apparent decrease of reactivity is discussed. Shortly thereafter (next par), the authors reveal that the reaction was carried on to a catecholic product. This is rather misleading. I suggest that the authors convey the full interpretation right away without 'mislead' readers first.

*We updated the text to be more straightforward regarding the catecholic product. Specifically, we have moved this part of the results section into the next subsection entitled "**Formation of a sequentially oxygenated product by Tsam**".*

(e12) p7, 1st par. The ability of Tsam and CAO to perform sequential monooxygenations is interesting but it might not apply to other ROs. Why would this comparison be particularly insightful?

Here, we suggest that this comparison is actually very interesting. There are ~70,000 annotated Rieske oxygenases and we do not know the function of a large majority of these enzymes. Here, we show an ability to introduce this sequential monooxygenation chemistry into Tsam, which is valuable because we learn about active site parameters that are potentially used by CAO to not only form the 7-hydroxymethyl intermediate, but also the formylated product (ACS Cent Sci. 2022, 8, 10, 1393–1403).

There is also a Rieske oxygenase PrnD, which is an N-oxygenase that catalyzes formation of an aryl nitro group. It was previously shown that sequential monooxygenation is the operative mechanism on a substrate analog (Angew. Chem. Int. Ed. 2006, 45 (4), 622-625), and therefore, presumably on the reported native substrate. These two examples hint that additional Rieske oxygenases that catalyze sequential monooxygenation reactions are yet to be discovered. We discuss only CAO in the text as it is arguably more relevant to Tsam chemistry.

(e13) p7, 1st par. Why should the acidity of C-H bonds matter? Wouldn't they be in the range of >15 and higher and far from 'acidic'?

This section has been substantially rewritten.

(e14) p7, 2nd par. This is difficult to read. If the whole purpose of this subsection was to start thinking about making sequential monooxygenases into dioxygenase, why not telling right away?

We rewrote this section and added the information from this paragraph into the following section entitled "Identification of architectural parameters involved in TsaM reactivity".

(e15) p7, 3rd par. 'Substrate electronics' is not a proper description of chemical properties and I feel that this hypothesis does not make sense. Even strong electron donating moieties of substrates do not necessarily make reactions with electrophilic Fe-oxygen go better or faster given that O₂ activation could be the overall rate-limiting step of catalysis.

We rewrote this section and removed this descriptor from the manuscript.

(e16) Figure 5a: Why did the authors not provide references to the corresponding papers? PDB ID are fine but access to the original numbers (in the cited papers) would be even more insightful to readers.

We thank Reviewer 2 for pointing out this omission. We added the required citations into the legends of Figure 5, Figure 6, and Supplementary Figure 44 for each of the structures that were used in our analysis.

(e17) p9, 3rd and 4th par. It would be interesting to have some visualization of the mutant enzymes active site structures for VanA and PDO. Could a reference to a modified Suppl. Fig. 38 help for this purpose?

We thank Reviewer 2 for this comment, we added an extra Supplementary Figure 64 to show a sequence alignment and an overlay of AlphaFold model of VanA with the X-ray structure of DdmC. This figure complements Supplementary Figure 50 which shows an overlay of the AlphaFold model of TsaM with the X-ray structure of DdmC. Last, we incorporated PDO structural insight into Figure 6b and Supplementary Figures 69 and 71.

Reviewer #3:

The manuscript by Tian et al. outlines the investigation of architectural features that dictate monooxygenation, sequential monooxygenation or dioxygenation reactions in the two-component Rieske oxygenase TsaM, and two additional representatives (VanA and PDO). The analysis was initiated by a thorough 'substrate engineering' approach comprising a wide range of substrates rationally chosen to characterize the catalytic ability of TsaM to catalyze monooxygenation, sequential monooxygenation, and even desaturation and dealkylation reactions. By performing rational protein design targeting several residues in the active site of TsaM, it was possible to identify several important amino acid residues that govern the catalytic outcome in this enzyme. Subsequently, the identified architectural features were transferred to VanA and PDO, two related Rieske's characterized as monooxygenase and dioxygenase, respectively. Evaluation of reaction outcome, created active site models and mutational studies were used to explain and validate findings and establish broader impact of the study.

The manuscript is scientifically sound and well-presented with a solid supporting information and methods part.

We thank Reviewer 3 for this positive assessment of our work, and we anticipate that that you will find the revised manuscript to be even more rigorous in the science and overall, more impactful to the community.

Major strengths and weaknesses of the present paper:

- The systematic analysis (and presentation in the manuscript) of the structural features that govern the reaction outcome in the protein family of Rieske oxygenases are positively noteworthy. The obtained results and interpretation thereof clearly contribute to increased knowledge and understanding of how these complex enzymes achieve different reaction outcomes (based on active site and substrate architecture).
- On the other hand, a demonstrated ability to predict or custom-tune monooxygenation / sequential monooxygenation over dioxygenation might not be fully supported by the study. The reason is that a full switch to monooxygenation in PDO by introducing the mutation I232G did not furnish the enzyme into an exclusive monooxygenase. Similarly, TsaM could not be fully engineered switched into a dioxygenase. As such, the herein-described principles of substrate architecture and proximity to the iron are accompanied by other factors that govern reaction outcomes that are of less 'predictive' nature.

We thank Reviewer 3 for this comment. We have now added four additional PDO variants (I256G/R207V, I256A/R207V, S182I/R207V, and I256G/S182I/R207V) to this work. Our data now shows that adding extra flexibility into the active site of PDO by combining the I256G construct with two extra mutations (R207V and S182I) that disrupt an interaction that anchors phthalate close to the iron, abolishes the ability of PDO to behave as a dearomatizing dioxygenase. Rather, a new triple variant (I256G/S182I/R207V) exclusively forms monooxygenated and sequentially monooxygenated products. This data can be found in Figure 6 and Supplementary Figure 71. Interestingly, the double variants I256G/R207V and I256A/R207V make approximately equal amounts of monooxygenated and dioxygenated products. Last, the PDO variant that lacks I256G is unable to perform monooxygenation chemistry at all.

Similarly, we made six additional variants of TsaM (M230W, T232W, T232I, M230W/T232I, M230W/T232I/S257R, and M230W/T232I/S257R/Y269V). We found that M230W and T232W variants lead to a similar amount of dioxygenated product formation as those previously reported (M230F and T232F). We further determined that a T232I variant (chosen as an intermediate size between T and F/W) forms a mixture of monooxygenated and dioxygenated products. Interestingly, we further determined that the M230W/T232I/S257R and M230W/T232I/S257R/Y269V variants show higher total turnover numbers with a benzoate substrate. Based on your comment below, we also tested if dioxygenated product formation could be amplified by addition of a non-native reductase to the reaction (the partner reductase protein of PDO). Using annotated reductase partners of PDO with the TsaM variants, the yield of the reaction approximately doubled (Figure 5c). We hypothesize, based on our most recently submitted manuscript, that the activity of this reaction might be further amplified by adjusting residues outside of the active site to recognize and position the substrate for catalysis.

- In addition, residues found within the active site of the alpha-subunit have been shown before to play key roles in dictating the site-, stereo-, and chemoselectivity of a Rieske oxygenase-

catalyzed reaction. For instance, it has been shown that in carbazole dioxygenase several mutations in the active site can change the orientation in which the substrate binds, and thus the outcome of the hydroxylation event (compare: Ref 20: Inoue K, et al. Appl Environ Microbiol 2014, 80:2821–2832).

Other studies of naphthalene dioxygenase also revealed that changes in the active site affect the reaction outcome. For instance, some form a monooxygenated product, rather than dioxygenated products using non-native substrates (compare: Ref. 54: Parales et al., J Bacteriol 2000, 182:1641–1649; Ref 57: Yu et al., J Ind Microbiol Biotechnol 2001, 27:94–103; Ref 32: Halder et al, ChemCatChem 2018, 10:178–182).

As such, the approach for studying the active site architecture and the reaction scope when using different non-natural substrates is based on these earlier studies. Conceptually, the ability to transfer mutations that influence reaction outcomes to closely related Rieske's is a novel aspect that strengthens the work.

We thank reviewer 3 for this comment. As described above, we added extra text into the introduction and discussion mentioning the works described above.

Other suggestions:

- The reported kinetic parameters (Table 1) for the different substrates require further clarification. Typically, k_{cat} values should be based on the number of active sites ($n=3?$) per trimer of the Oxy. The authors should clarify whether if this has been taken into account and indicate this in the methods part.

We thank Reviewer 3 for this comment. To correct this omission, we added a clarification into the methods section regarding how we determined the reported values in table. At the end of the related section, we added a statement that reads "In all cases, since each monomeric unit of a Rieske oxygenase contains one active site, the k_{cat} was calculated using the concentration of a single protomer." We added a similar statement regarding how we calculated the total turnover numbers.

- In the a_3 architecture such as in TsaM, the subunit-subunit interfaces are the site where the partner reductase bind. For TsaM as well as for PDO, the 'non-native' redox partner VanB has been used. Since the binding of the redox partner induces several conformational changes to facilitate electron transfer, a non-native redox partner might also influence the reaction outcome. As for TsaM the native reductase TsaB could not be functionally reconstituted in vitro, I suggest performing a control experiment with PDO and its native reductase PDR for the wild-type PDO and the variant I256G to exclude an effect of the non-native redox partner on the reaction outcome.

*We thank Reviewer 3 for this comment. We now added extra experiments that show the TsaM-catalyzed dioxygenation reactions can also be catalyzed with TsaB. The activity of these assays, as with the reported native substrates is quite low. Therefore, we only observe formation of the dioxygenated product with the most active TsaM variants (M230W/T232I/S257R and M230W/T232I/S257R/Y269V). This data is included in Supplementary Figure 54. As described above, however, using the annotated reductase partners of PDO from *C. testosteroni* (PDR1) or *Pseudomonas cepacia* (PDR2), we demonstrate that the yield of the TsaM-catalyzed dioxygenation reaction can be amplified. This data is now included in Figure 5c.*

In addition, we added experiments to show that the combination of PDO (or PDO variants) with the two different annotated PDO reductase proteins (PDR1 and PDR2) leads to increased formation of the same products observed using the non-native (VanB) reductase. This data is included in Supplementary Figures 69-71.

- Another point is the dependency of oxygen on the reaction outcome (mono- vs. dihydroxylation). Did the authors observe any effect of oxygen on the reaction outcome, e.g. by determining the apparent K_M for oxygen?

We did not determine the K_M for oxygen but have now made it clear that we are reporting apparent kinetic parameters in this manuscript. With the added uncoupling experiments, we assert that these further experiments are important future work but outside the scope of this article.

REVIEWER COMMENTS

Reviewer #1 (Remarks to the Author):

I am very grateful for the author's thoughtful and detailed responses, which solved most of my questions. However, I am still doubtful about the sequential monooxygenation proposed by the authors, because the data in Figures 4d and 5d was not convincing.

The experiments were performed with 400 μM substrates and 2.5 μM TsaM (2, 5, or 10 μM ?). Only 10 - 20 % of substrates (benzenesulfonate or benzoate) were oxidized by TsaM according to the concentrations of products, and about 90% of substrates ($\sim 300 - 350 \mu\text{M}$) were thought to remain. Here comes the question: why did TsaM oxidize the monooxygenated products (4-hydroxybenzenesulfonate or 4-hydroxybenzoate) with concentration (less than $40\mu\text{M}$) one order of magnitude lower than the substrates (about $300-350 \mu\text{M}$), but not the latter? It should have only $\sim 10\%$ and 90% chance to act on those monooxygenated products and the original substrate, respectively, judged from the concentrations and the K_m values. Then, we should observe the increasing concentration of the monooxygenated products until its concentration reached comparable to that of the remained substrates. After that, the concentrations of dioxygenated products would significantly increase. Now I have another question, did the rest of the substrates still remain? Or have those substrates been oxidized to other species? A mass balance assay of consumed substrates and yield products was suggested.

Reviewer #2 (Remarks to the Author):

Tian et al. show through targeted mutagenesis experiments that a rational design of the active site space of Rieske oxygenases allows guiding the reaction outcome from monooxygenations to sequential monooxygenations, and, finally, dioxygenations. The authors make their case by confirming the hypothesis that the positioning of substrate towards the reactive non-heme Fe center, specifically the distance between Fe and the hydroxylated C atom(s) of the substrate, largely determines the reaction outcome.

As a response to the various comments of three reviewers, the authors have added additional data to an already comprehensive study and extended their discussion toward additional factors that determine the reactivity of Rieske oxygenases.

I have reviewed the original submission as reviewer no. 2 and commend the authors for elaborating on the reviewer suggestions very carefully. My principal suggestions regarding the role of productive/unproductive activation of dioxygen and interpretation of kinetic parameters have been largely addressed. In my opinion, the revised manuscript could benefit from some additional editorial revisions to clarify these points a bit more.

I am curious to follow the discussions in the scientific community triggered by this work.

General comments

(1) The authors now provide quantitative estimates for unproductive O₂ activation through quantification of H₂O₂ with the Amplex Red assays as well as qualitatively through searches for oxidative modification of the oxygenase through intact protein mass spectrometry. These data are valuable additions to the manuscript and support the overall conclusion of this work well.

Nevertheless, I would recommend that the authors added a statement, for example, in the Methods section, that the O₂ uncoupling assays provide a lower estimate of the extent of O₂ uncoupling. Not all of the unproductively activated O₂ winds up in H₂O₂. Moreover, the accuracy of H₂O₂ assays in matrices used for studying Rieske oxygenases is, unfortunately, far from perfect. And the data from intact protein mass spectrometry lack validation through positive controls. It is far beyond the work of Tian et al. to elaborate on these phenomena and I do not request any additional data with this comment. But a short disclaimer on the “accuracy” of insights from the uncoupling assays would add to the already excellent scientific quality of this work.

(2) Parts of the study’s approach does not become fully clear in the introduction. While this work focuses explicitly on oxygenation reactions of toluenesulfonate methyl-monooxygenase (TsaM), the experimental and theoretical evaluation of vanillate O-demethylase (VanA), phthalate dioxygenase (PDO), and dicamba monooxygenase (DdmC) also contribute critically to the conclusions made. I wonder whether the authors could add a sentence on page 3 in the introduction to make readers aware of the comprehensive (multi-enzyme) approach taken.

Editorial revisions

p 2 / par 1: “metallocenters must be reduced”. Replace with „iron atoms in both the Rieske cluster and non-heme iron site must be in its reduced state“.

p 2 / par 1: It would be sufficient to refer to electron shuttling across the subunit interface instead of “subunit-subunit” interface.

p 3 / par 2: unclear meaning of “compounds that are easily accessed.”

Throughout ms: Replace/delete colloquial terms related to “dearomatization” such as “dearomatizing dioxygenation” etc. with chemically accurate terminology. Hydroxylation of aromatic (and olefinic) carbon includes a change carbon hybridization but this is not an a priori outcome of catalysis by Rieske oxygenases. Some dihydroxylated substrates spontaneously rearrange to regain aromaticity. Referring to dioxygenation without “dearomatization” would be fully sufficient. Instead of dearomatized cis-diols (e.g., page 14), I suggest to use “cis-dihydrodiols”. This expression adequately implies change of carbon hybridization.

Reviewer #3 (Remarks to the Author):

I have read both present and previous versions of the manuscripts and the authors' responses to reviewers. The authors carefully revised the manuscript according to the Reviewers' comments. The authors' responses are reasonable, and with the additional data and the changes to the main manuscript, the quality of the manuscript is now excellent. Thus, I would support publication as is at this point.

Reviewer #1 (Remarks to the Author):

I am very grateful for the author's thoughtful and detailed responses, which solved most of my questions. However, I am still doubtful about the sequential monooxygenation proposed by the authors, because the data in Figures 4d and 5d was not convincing. The experiments were performed with 400 μM substrates and 2.5 μM TsaM (2, 5, or 10 μM ?). Only 10 - 20 % of substrates (benzenesulfonate or benzoate) were oxidized by TsaM according to the concentrations of products, and about 90% of substrates ($\sim 300 - 350 \mu\text{M}$) were thought to remain.

Here comes the question: why did TsaM oxidize the monooxygenated products (4-hydroxybenzenesulfonate or 4-hydroxybenzoate) with concentration (less than 40 μM) one order of magnitude lower than the substrates (about 300-350 μM), but not the latter? It should have only ~ 10 % and 90% chance to act on those monooxygenated products and the original substrate, respectively, judged from the concentrations and the K_m values. Then, we should observe the increasing concentration of the monooxygenated products until its concentration reached comparable to that of the remained substrates. After that, the concentrations of dioxygenated products would significantly increase.

Now I have another question, did the rest of the substrates still remain? Or have those substrates been oxidized to other species? A mass balance assay of consumed substrates and yield products was suggested.

We thank Reviewer 1 for the clarifications made to the previous comment. To address your question, we have now included the mass balance experiment. You can find this experiment as part of Supplementary Figures 36 (benzenesulfonate) and 37 (benzoate). This experiment, as you suggest, is quite interesting because we see the appearance of the monooxygenated (4-hydroxybenzenesulfonate and 4-hydroxybenzoate) products early and then the dioxygenated products (3,4-dihydroxybenzenesulfonate and 3,4-dihydroxybenzoate) form. Even after 3,4-dihydroxybenzenesulfonate and 3,4-dihydroxybenzoate products are detected, we don't observe additional formation of the monooxygenated species. Rather, we find that increasing concentrations of 3,4-dihydroxybenzenesulfonate and 3,4-dihydroxybenzoate correlate with decreasing concentrations of 4-hydroxybenzenesulfonate and 4-hydroxybenzoate, a trend that is consistent with sequential monooxygenation reactions. To further investigate this phenomenon, we measured NADH consumption using a UV-Vis assay. This assay was included because, as discussed in the last round of revisions with Reviewer 2, the Michaelis Menten kinetic parameters "reflect the rates of O_2 activation in the presence of a specific substrate than the hydroxylation of its aromatic or aliphatic C atoms". In this experiment, we determined that NADH is consumed at a faster rate in the presence of 4-hydroxybenzenesulfonate and 4-hydroxybenzoate relative to benzenesulfonate and benzoate (Supplementary Figure 38). With respect to question raised by Reviewer 1, "why did TsaM oxidize the monooxygenated products", we hypothesize that this data supports the assertion that the monooxygenated species more closely resemble the native substrates and therefore adopt "near-native" binding poses in the active site of TsaM. In contrast the smaller benzenesulfonate and benzoate substrates likely adopt ensembles of binding poses that do not facilitate downstream catalytic processes.

*We also added in the mass balance experiment for the M230G/T232G variant of TsaM that makes 4-formylbenzenesulfonate and 4-formylbenzoate. This experiment can be found in Supplementary Figure 63 (*p*-toluenesulfonate and *p*-methylbenzoate substrates). Here, we see a similar trend to that described for benzenesulfonate and benzoate, but the activity on the starting molecule is higher.*

*With respect to the second question raised by Reviewer 1, for all substrates (benzenesulfonate, benzoate, *p*-toluenesulfonate, and *p*-methylbenzoate), the starting concentration of the substrate is fully accounted for at the end of the reaction. We added some explanation of these phenomena into the discussion (see next paragraphs) and to support this added text, an additional bar graph and DFT calculations were added to Supplementary Figures 34 and 38, respectively.*

*"An analogous hypothesis can also be applied to the experiments performed with benzenesulfonate and benzoate substrates, which are both smaller than *p*-toluenesulfonate and 4-methylbenzoate. In fact, in the sequential monooxygenation reaction, it is quite remarkable that the products of the first oxygenation event, 4-hydroxybenzenesulfonate and 4-hydroxybenzoate, are oxygenated in the presence of a large excess of benzenesulfonate and benzoate. We attribute this phenomenon to the size of the monooxygenated compounds which more closely resemble *p*-toluenesulfonate and 4-methylbenzoate and likely supports a near-native binding orientation in the active site (Supplementary Fig. 34). Similarly, it is worth mentioning that relative to benzoate, the partial charge distribution of 4-hydroxybenzoate more closely approximates *p*-(methoxy)benzoate, a molecule that is turned over to a significantly greater extent than benzoate by TsaM (Supplementary Fig. 34 and*

38f). This latter point again suggests that a higher proportion of the monooxygenated compounds will adopt productive binding orientations relative to that of benzenesulfonate and benzoate in the TsaM active site. Therefore, the monooxygenated compounds are expected to better support the needed downstream processes for catalysis. Perhaps related to this sentiment, here it was shown that NADH consumption is increased in the presence of 4-hydroxybenzoate and 4-hydroxybenzenesulfonate relative to their non-oxygenated counterparts (Supplementary Fig. 38).

Similar parallels can be made for the tested TsaM variants (M230F/W, T232F/W, M230W/T232I, M230W/T232I/S257R, or M230W/T232I/S257R/Y269V). Here, the active sites are engineered to confer dioxygenation chemistry, but the protein scaffold is not optimized for a benzoate substrate. As such, some of the benzoate molecules likely adopt binding orientations that are productive, and others adopt poses that negatively impact the protein behaviors coupled to catalysis (Figure 5c). In contrast, the non-native sequential monooxygenation and monooxygenation activity of TsaM and PDO variants is generally higher in the presence of the reported native substrates (Figure 5d, Figure 6b, and Supplementary Fig. 69-71). In particular for the M230G/T232G TsaM variant, it is interesting to note that the polar charge distributions at the *p*-positions of the reported native products, *p*-(hydroxymethyl)benzenesulfonate and 4-(hydroxymethyl)benzoate, resemble the polar charge distributions at the *p*-positions of 4-hydroxybenzenesulfonate and 4-hydroxybenzoate (Supplementary Fig. 34). However, *p*-(hydroxymethyl)benzenesulfonate and 4-(hydroxymethyl)benzoate have added steric bulk conferred by the additional methylene group. This extra bulk is likely the culprit that prevents wild-type TsaM from catalyzing a CAO-like second reaction on these molecules, but it is intriguing that the introduction of two active site Gly residues, a presumable extension of the active site ruler, does permit sequential monooxygenation reactions on *p*-toluenesulfonate and 4-methylbenzoate (Figure 5d and Supplementary Fig. 40). Collectively, these results suggest that combination of a native substrate with an altered enzyme active site permits better control over downstream catalytic behaviors, including O₂ binding, electron transfer, and a productive interaction with the reductase^{30,31,77}. Such an interaction has been previously shown to induce conformational changes in the Rieske oxygenase, and therefore, could presumably also influence reaction outcome^{30,31}.

For the benefit of Reviewer 1 (and for review only), we also wanted to say that during this review process, we submitted a manuscript about the enzymes TsaC and TsaD, which function downstream of TsaM/TsaB. The annotated functions of these proteins are to sequentially convert the alcohol product of TsaM/TsaB into an aldehyde and carboxylic acid, respectively. To provide additional support that the aldehyde is indeed being generated by the M230G/T232G variant of TsaM, we purified the enzyme TsaD, which is annotated to convert both 4-formylbenzenesulfonate and 4-formylbenzoate into the carboxylic acid containing compounds, 4-sulfobenzoate and 1,4-benzenedicarboxylate, respectively. In the data below, you will see that when TsaD is included in the assay with M230G/T232G TsaM, VanB, NADH and 4-methylbenzenesulfonate or 4-methylbenzoate, we see formation of the corresponding alcohol, aldehyde, and carboxylic acid products.

Figure 1. TsaD converts the aldehyde produced by the the M230G/T232G variant of TsaM into carboxylic acid-containing products. (a) Time dependent formation of 4-sulfobenzoate in reactions that contain TsaM, VanB, NADH, p-toluenesulfonate, and TsaD. (b) Time dependent formation of 1,4-benzenedicarboxylate in reactions that contain TsaM, VanB, NADH, 4-methylbenzoate, and TsaD. (c) A control reaction for panel a that lacks TsaD shows no formation of 4-sulfobenzoate. (d) A control reaction for panel a that lacks TsaD shows no formation of 1,4-benzenedicarboxylate. In this figure, for panels a-b, [TsaM] = 10 μ M, [VanB] = 40 μ M, [TsaD] = 10 μ M, and [initial substrate] = 1 mM. For panels c-d [TsaM] = 2 μ M, [VanB] = 8 μ M, [TsaD] = 0 μ M, and [initial substrate] = 0.4 mM.

Reviewer #2 (Remarks to the Author):

Tian et al. show through targeted mutagenesis experiments that a rational design of the active site space of Rieske oxygenases allows guiding the reaction outcome from monooxygenations to sequential monooxygenations, and, finally, dioxygenations. The authors make their case by confirming the hypothesis that the positioning of substrate towards the reactive non-heme Fe center, specifically the distance between Fe and the hydroxylated C atom(s) of the substrate, largely determines the reaction outcome.

As a response to the various comments of three reviewers, the authors have added additional data to an already comprehensive study and extended their discussion toward additional factors that determine the reactivity of Rieske oxygenases.

I have reviewed the original submission as reviewer no. 2 and commend the authors for elaborating on the reviewer suggestions very carefully. My principal suggestions regarding the role of productive/unproductive activation of dioxygen and interpretation of kinetic parameters have been largely addressed. In my opinion, the revised manuscript could benefit from some additional editorial revisions to clarify these points a bit more.

I am curious to follow the discussions in the scientific community triggered by this work.

General comments

(1) The authors now provide quantitative estimates for unproductive O₂ activation through quantification of H₂O₂ with the Amplex Red assays as well as qualitatively through searches for oxidative modification of the oxygenase through intact protein mass spectrometry. These data are valuable additions to the manuscript and support the overall conclusion of this work well.

Nevertheless, I would recommend that the authors added a statement, for example, in the Methods section, that the O₂ uncoupling assays provide a lower estimate of the extent of O₂ uncoupling. Not all of the unproductively activated O₂ winds up in H₂O₂. Moreover, the accuracy of H₂O₂ assays in matrices used for studying Rieske oxygenases is, unfortunately, far from perfect. And the data from intact protein mass spectrometry lack validation through positive controls. It is far beyond the work of Tian et al. to elaborate on these phenomena and I do not request any additional data with this comment. But a short disclaimer on the “accuracy” of insights from the uncoupling assays would add to the already excellent scientific quality of this work.

We have updated the methods (now included in the main text document) to contain two additional statements:

Statement 1: To accomplish these measurements and obtain an estimate of the extent of O₂ uncoupling in our reactions, we capitalized on the availability of an Invitrogen Amplex Red hydrogen peroxide/peroxidase assay kit.

Statement 2: Of note, this experiment exclusively provides a lower estimate of the unproductively activated O₂ that is lost as H₂O₂.

(2) Parts of the study’s approach does not become fully clear in the introduction. While this work focuses explicitly on oxygenation reactions of toluenesulfonate methyl-monooxygenase (TsaM), the experimental and theoretical evaluation of vanillate O-demethylase (VanA), phthalate dioxygenase (PDO), and dicamba monooxygenase (DdmC) also contribute critically to the conclusions made. I wonder whether the authors could add a sentence on page 3 in the introduction to make readers aware of the comprehensive (multi-enzyme) approach taken.

We updated the text on page 3 to include two additional statements:

Statement 1: By capitalizing on these substrate recognition elements and available Rieske oxygenase structural information...

Statement 2: Complementary work on vanillate O-demethylase (VanA) and phthalate dioxygenase (PDO) revealed that this rational strategy can be used to similarly manipulate the reaction outcome of other Rieske oxygenase enzymes: VanA was engineered to perform dioxygenation chemistry and PDO was engineered to function as a monooxygenase.

Editorial revisions

p 2 / par 1: “metallocenters must be reduced”. Replace with „iron atoms in both the Rieske cluster and non-heme iron site must be in its reduced state“.

We updated the text as suggested.

p 2 / par 1: It would be sufficient to refer to electron shuttling across the subunit interface instead of “subunit-subunit” interface.

We updated the text as suggested.

p 3 / par 2: unclear meaning of “compounds that are easily accessed”.

We updated the text to read “compounds that can be readily purchased”.

Throughout ms: Replace/delete colloquial terms related to “dearomatization” such as “dearomatizing dioxygenation” etc. with chemically accurate terminology. Hydroxylation of aromatic (and olefinic) carbon includes a change carbon hybridization but this is not an a priori outcome of catalysis by Rieske oxygenases. Some dihydroxylated substrates spontaneously rearrange to regain aromaticity. Referring to dioxygenation without “dearomatization” would be fully sufficient. Instead of dearomatized cis-diols (e.g., page 14), I suggest to use “cis-dihydrodiols”. This expression adequately implies change of carbon hybridization.

We updated the text as suggested.

Reviewer #3 (Remarks to the Author):

I have read both present and previous versions of the manuscripts and the authors' responses to reviewers. The authors carefully revised the manuscript according to the Reviewers' comments. The authors' responses are reasonable, and with the additional data and the changes to the main manuscript, the quality of the manuscript is now excellent. Thus, I would support publication as is at this point.

We thank Reviewer 3 for this comment and support of this publication.

REVIEWERS' COMMENTS

Reviewer #1 (Remarks to the Author):

The authors' effort are deeply appreciated. The mass balance assay confirmed the production of the monoxygenated and dioxygenated species. And I do not doubt this now. However, the question still remained why Tsam and variants preferentially oxidized the monoxygenated products (such as 4-hydroxybenzenesulfonate, compound 41) under the situation where the concentrations of monoxygenated products ($< 40 \mu\text{M}$) were significantly lower than initial substrates (such as benzenesulfonate, compound 37, $\sim 300 \mu\text{M}$). Compounds 37 and 41 had the similar K_M , K_{cat} and V_{max} (Table 1), which suggested Tsam should preferentially oxidize those compounds with a higher concentration (compound 37). Yet the data in Figure 4 and 5 indicated Tsam preferentially oxidize those compounds with a lower concentration (compound 41). They seemed contradict each other. Actually I could not propose a reasonable hypothesis about that for this moment and think the authors should discuss that.

Reviewer #1 (Remarks to the Author):

The authors' effort are deeply appreciated. The mass balance assay confirmed the production of the monooxygenated and dioxygenated species. And I do not doubt this now. However, the question still remained why TsaM and variants preferentially oxidized the monooxygenated products (such as 4-hydroxybenzenesulfonate, compound 41) under the situation where the concentrations of monooxygenated products ($< 40 \mu\text{M}$) were significantly lower than initial substrates (such as benzenesulfonate, compound 37, $\sim 300 \mu\text{M}$). Compounds 37 and 41 had the similar K_M , K_{cat} and V_{max} (Table 1), which suggested TsaM should preferentially oxidize those compounds with a higher concentration (compound 37). Yet the data in Figure 4 and 5 indicated TsaM preferentially oxidize those compounds with a lower concentration (compound 41). They seemed contradict each other. Actually I could not propose a reasonable hypothesis about that for this moment and think the authors should discuss that.

We thank reviewer 1 for recognition of our efforts to revise the paper. We extended the discussion of this phenomenon. Specifically, we added the following information:

Quite interestingly, the kinetic parameters for each of these different molecules are nearly equivalent (Table 1). The basis of this phenomenon, at this point, is unclear. Typically, this finding would suggest that the rate of O_2 activation in the presence of the different substrate options is similar. However, we anticipate that the values measured for this set of substrates are not directly comparable due to the sequential nature of the reaction. Rather, as described above, it is possible that the chemical attributes of these molecules allow for an ensemble of binding poses in the protein active site. The proportion of these molecules that are positioned correctly permit catalysis and immediately produce either 4-hydroxybenzenesulfonate and 4-hydroxybenzoate, which are also substrates of the reaction. These measured kinetic parameters with each these different molecules will be the basis of future investigation. However, we still posit that the monooxygenated compounds are expected to better bind in the active site and support the needed downstream processes for catalysis. Perhaps related to this interpretation, here it is shown that NADH consumption is increased in the presence of 4-hydroxybenzoate and 4-hydroxybenzenesulfonate relative to their non-oxygenated counterparts (Supplementary Fig. 38).